

# A new 2010 permafrost distribution map over the Qinghai-Tibet Plateau based on subregion survey maps: a benchmark for regional permafrost modeling

Zetao Cao[1], Zhuotong Nan[1,2,*], Jianan Hu[1], Yuhong Chen[1], Yaonan Zhang[3]

[1]Key Laboratory of Ministry of Education on Virtual Geographic Environment, Nanjing Normal University, Nanjing 210023, China;
[2]Jiangsu Center for Collaborative Innovation in Geographical Information Resource Development and Application, Nanjing 210023, China;
[3]National Cryosphere Desert Data Center, Lanzhou 730000, China

*Correspondence to*: Zhuotong Nan (nanzt@njnu.edu.cn)

**Abstract.** Permafrost over the Qinghai-Tibet Plateau (QTP) has gained increasing attention due to its high sensitivity to climate change. Numerous spatial modeling studies have been conducted on the QTP to assess the status of permafrost, project future changes in permafrost, and diagnose contributors to permafrost degradation. Due to very limited number of observation stations on the QTP, these modeling studies are often hampered by the lack of validation references, calibration targets, and model constraints, whereas a high-quality permafrost distribution map can be a good option as a benchmark for spatial simulation results. Existing permafrost distribution maps on the QTP can hardly serve this purpose. An ideal benchmark map for spatial modeling should be methodologically sound, have sufficient accuracy, and be based on observations collected in specific mapping years, rather than all historical data spanning several decades. Therefore, in this study, we created a new permafrost distribution map over the QTP in 2010 through a novel permafrost mapping approach with satellite-derived ground surface thawing/freezing indices as input and survey-based subregion permafrost maps as constraints. This approach was further improved in this study to reduce parametric equifinality. It accounts for the effects of local factors by incorporating into the model an empirical soil parameter whose values are optimally estimated through spatial clustering and parameter optimization constrained by survey-based subregion permafrost maps. This new map shows a total permafrost area of about $1.086 \times 10^6$ km$^2$ (41.2% of the QTP area) and seasonally frozen ground of about $1.447 \times 10^6$ km$^2$ (54.9%) in 2010, excluding glaciers and lakes. Validations using survey-based subregion permafrost maps (Kappa = 0.74) and borehole records (Overall Accuracy = 0.85 and Kappa = 0.43) showed higher accuracy of this map than two recent maps. Inspection of regions with obvious distinctions between the maps affirms that the permafrost distribution on this map is more realistic than on the Zou et al. (2017) map. Due to the excellent accuracy demonstrated, this map can serve as a benchmark map of sufficient quality to constrain/validate land surface models and as a historical reference when projecting future permafrost changes on the QTP in the context of global warming.





## 1 Introduction

Permafrost, defined as ground that remains at or below 0 °C for at least two consecutive years (Dobinski, 2011), underlies more than 20% of the exposed land area in the Northern Hemisphere (Obu et al., 2019) and constitutes an essential

component of the Earth system. As such, studies on permafrost have attracted increasing interest for its sensitive and complex responses and feedbacks to global climate change (Slater and Lawrence, 2013; Schuur et al., 2015; Walvoord and Kurylyk, 2016). The Qinghai-Tibet Plateau (QTP), also known as the Earth's third pole, has the highest and largest low-latitude permafrost in the world (Cao et al., 2019a). Affected by the complex topography and unique plateau climate, permafrost over the QTP is of low thermal stability in general and strongly influenced by complex local factors, thus

differing from high-latitude permafrost around the Arctic and being more sensitive to global climate change (Li et al., 2008; Yang et al., 2019; Zhao et al., 2020).

In the context of global warming, significant permafrost degradation occurs on the QTP and has a great impact on hydrological processes (Li et al., 2020), carbon cycle (Mu et al., 2020), and thermodynamic processes (Zhao et al., 2020). Hazards related to permafrost degradation threaten constructions and infrastructures on the QTP (Wang et al., 2020).

Meanwhile, the consequences will lead to vital feedbacks to climate systems (Zhang et al., 2020; Wang et al., 2021). Many researches have studied the complex responses and feedbacks of permafrost to climate change (Yang et al., 2019), while spatial modeling of permafrost dynamics using land surface models has become an important approach (Ji et al., 2022). Using land surface models, many spatial modeling studies have attempted to project future changes in permafrost distribution(Chang et al., 2018; Debolskiy et al., 2020; Yin et al., 2021), assess permafrost physics under climate change

(Koven et al., 2013; Burke et al., 2020), diagnose the contributors to regional permafrost degradation (Zhang et al., 2021; Zhang et al., 2021; Mekonnen et al., 2021), and project possible feedbacks to climate systems due to permafrost degradation (Yokohata et al., 2020; Andresen et al., 2020). But the evaluation of spatial modeling results has often been a major challenge for these studies. Compared with the large extent of spatial modeling results, field observations are often too sparse to provide references, especially on the QTP. In such circumstances, using an accurate permafrost distribution map as

a target to calibrate the parameters and as a reference to validate the results can be a good choice. More importantly, a historical map of permafrost distribution can provide a constraint for future projection studies to minimize biases arising from the modeling process. Moreover, an accurate permafrost distribution map can serve as a fundamental dataset for hydrological, carbon, ecological and engineering studies in cold regions (Hu et al., 2019; Li et al., 2020; Song et al., 2020; Mu et al., 2020).

Though many permafrost distribution maps have been compiled over the QTP (Cheng et al., 2011; Shi and Mi, 2013; Wang, 2013; Guo and Wang, 2013; Zou et al., 2017; Niu and Yin, 2018; Shi et al., 2018; Wu et al., 2018; Wang et al., 2019), the quality of these maps is often unsatisfactory, and few of them can serve as a valid benchmark for calibrating and validating land surface models (Wang et al., 2016; Cao et al., 2019a). The reasons restricting the accuracy and reliability of permafrost distribution maps on the QTP are mainly due to insufficient data used in compilation and inadequate mapping





approaches. Early permafrost maps on the QTP (Cheng et al., 2011; Shi and Mi, 2013) were compiled through visual interpretation based on limited data and expert judgment. Subsequently, satellite data and reanalysis data have become the main data sources of permafrost mapping (Wang, 2013; Zou et al., 2017; Shi et al., 2018; Wang et al., 2019). But large gaps in satellite data caused by clouds would highly affect the accuracy of permafrost maps in absence of effective interpolation methods (Chen et al., 2020). Although the reanalysis products do not suffer from cloud contamination, their coarse spatial

resolutions and unsatisfactory performances on the QTP (Hu et al., 2019; Qin et al., 2020; Cao et al., 2020) will consequently restrict the accuracy of the derived maps.

Uncertainties related to mapping approaches also negatively exert influence on the performance of existing permafrost maps. Currently, popular statistical learning methods applied to permafrost mapping (Wang et al., 2019; Ni et al., 2021) are often compromised by the uneven distribution of field observation sites spanning different observation periods. It will

consequently lead to misrepresentation and overfitting in permafrost maps (Marcer et al., 2017), as these statistical learning methods rely heavily on in situ observations as a training dataset. Meanwhile, the lack of detailed soil properties, adequate parameterization schemes, and high-resolution forcing data on the QTP severely challenge the performances of land surface models in mapping permafrost (Wu et al., 2018). These physically-explicit models were often calibrated and validated at a point scale, leading to unpredictable uncertainties when extended to a large region with complex conditions. In addition,

permafrost distribution results generated by land surface models are usually difficult to serve as an independent benchmark due to the more or less similarity among them in model structure and forcing data. Therefore, empirical and semi-physical approaches remain the main approaches in permafrost mapping on the QTP because of their lower need for in situ observations than statistical learning methods and simpler structure with fewer parameters than physical models (Zou et al., 2017; Zhao et al., 2017). Nevertheless, these maps are criticized for limited consideration of local-scale permafrost control

factors (Cao et al., 2019a; Hu et al., 2020), and no constraints set up during simulation to avoid divergence. All these issues call into question the ability of the existing permafrost distribution maps to serve as a baseline or benchmark with sufficiently high spatial resolution and quality for land surface models and climate models on the QTP.

An ideal benchmark map for spatial modeling of permafrost should be, firstly, a map based on credible observations and independent of land surface model simulations; secondly, a map based on multi-year data close to a specific mapping

year, rather than all historical data spanning decades, thereby allowing it as a calibration target or a constraint for transient models; last but not least, a map of adequate accuracy by considering the impacts of local factors and well constrained during the mapping process. Under such circumstances, this study aims to provide a high-quality permafrost distribution map over the QTP in 2010 through an effective permafrost mapping approach that takes into account the effects of local factors and fully utilizes observational data, including remote sensing data and survey-based subregion permafrost maps, as an

optimization and constraint target. This new permafrost distribution map over the QTP can serve as a new reference map of 2010 for regional permafrost simulation studies and provide the benchmark for transient land surface models under climate change.



## 2 Data

### 2.1 Subregion permafrost maps

From 2009 to 2014, a research project sponsored by China Minister of Science and Technology was carried out to investigate permafrost and its surroundings. Intensive surveys were conducted in five representative areas (i.e. West Kunlun, Gaize, Aerjin, National Highway G308, and Wenquan, west to east in Fig.1) with distinct climatic and geographic conditions (Zhao et al., 2017). Thanks to comprehensive information provided by field observations, mechanical excavation, geophysical exploration techniques (ground penetrating radar, time-domain electromagnetic) and borehole drilling, the

permafrost distributions were mapped with unprecedented quality in all five subregions. These subregion permafrost maps have been widely used as ground truth for assessment in many studies (Zou et al., 2017; Zhao et al., 2017; Shi et al., 2018; Wu et al., 2018; Wang et al., 2019). In this study, the permafrost maps in the five subregions were used as optimization targets to calibrate a model soil parameter representing a synthesized soil thermal and moisture condition.

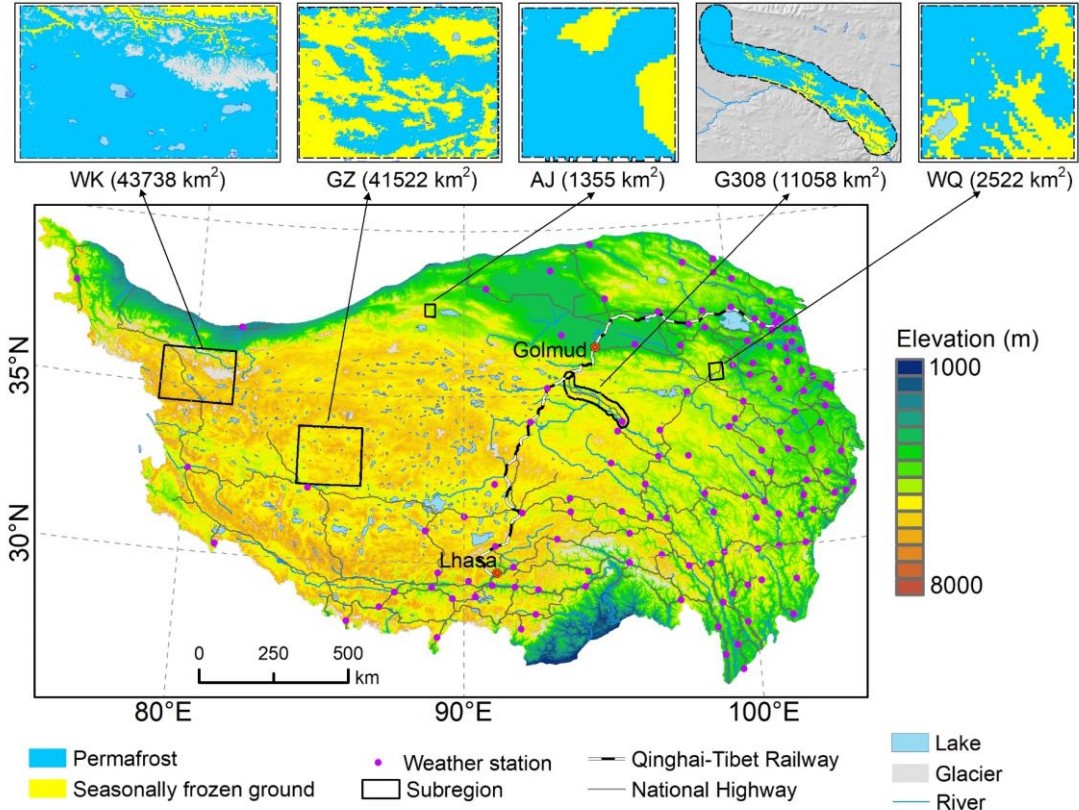

**Figure 1. Map showing the topography of the Qinghai-Tibet Plateau (QTP), the locations of meteorological stations, and the subregions with extensive field investigation. Inset maps show the local permafrost distributions in the five subregions (West Kunlun, Gaize, Aerjin, G308, and Wenquan, from west to east) based on the survey data circa 2010. WK: West Kunlun; GZ: Gaize; AJ: Aerjin; WQ: Wenquan (WQ).**



## 2.2 Satellite land surface temperature product

The land surface temperature (LST) product from the Moderate Resolution Imaging Spectroradiometer (MODIS) onboard the Terra and Aqua satellites is one of the most widely used LST products due to its high spatial and temporal resolutions and global coverage (Wan, 2008) and has been applied in many permafrost mapping studies to provide temperature conditions (Gisnås et al., 2017; Zou et al., 2017; Obu et al., 2019; Wang et al., 2019). In this study, we employed daily MODIS LST/emissivity products (MOD11A1 and MYD11A1 version 6), which provide up to two daytime

and two nighttime LST observations per day at 1 km resolution. These data are used to estimate ground surface thawing indices (DDT) and freezing indices (DDF), which in theory are aggregated from the 0 cm ground temperature (herein ground surface temperature, GST), as drivers to the mapping approach.

Moreover, to avoid the influence of single-year meteorological anomalies, it is necessary to calculate multi-year averages of DDT and DDF. Since the meteorological variables were measured manually until 2005, when automatic weather

stations were put into operation over the study area, we used MODIS LST products from 2005 to 2010 accordingly.

## 2.3 Environmental factors related to permafrost distribution

To account for the influence of local factors on permafrost distribution, several environmental factors were included in the mapping approach. The composite 16-day 1-km normalized difference vegetation index (NDVI) product (MOD13A2) provides the vegetation conditions over the study area. These data from 2005 to 2010 were aggregated into a multi-year

average NDVI. They were also used as an auxiliary variable in the estimation of DDT from satellite LST data.

Topographical factors, including elevation and slope, were derived from the Shuttle Radar Topography Mission 90m digital elevation database (SRTM/DEM, version 4) (Reuter et al., 2007), and then resampled to a spatial resolution of 1 km. The STRM/DEM-derived topographic wetness index (TWI) resampled to 1 km, together with the 2005 to 2010 mean annual precipitation aggregated from the 1-km monthly precipitation dataset for China (Peng et al., 2019), represent the wetness

condition affecting permafrost distribution. In addition, fraction snow cover (FSC) data for the same period from the 500 m Daily Fractional Snow Cover Dataset Over High Asia (Qiu et al., 2017) were aggregated into a multi-year average and resampled to 1 km as one of the environmental factors related to the ground thermal regime. Soil texture type data derived from the China Data Set of Soil Properties for Land Surface Modeling (Shangguan et al., 2013) were used in the mapping approach.

## 2.4 In situ observations

### 2.4.1 Ground surface temperature (GST) observations

There are 131 national meteorological stations of China on the QTP (Fig. 1), which are mostly concentrated in the eastern QTP. At these stations, standard meteorological variables are measured 4 times a day, at 2, 8, 14, and 20 o'clock. We extracted the daily GST observations during the period from 2005 to 2010 at these stations from the daily meteorological



dataset of basic meteorological elements of the China national surface weather stations (version 3.0) (National Meteorological Information Center, 2019). The in situ GST observations were used as a reference for the estimates of DDT and DDF on the QTP.

### 2.4.2 Permafrost presence/absence observations

We collected permafrost presence/absence information revealed by boreholes to evaluate the new permafrost
distribution map of this study. A newly published synthesis dataset of permafrost thermal state on the QTP (Zhao et al., 2021) containing soil temperatures measured at 10m and 20m depths from 2005 to 2018 was acquired to determine the presence of permafrost at borehole locations based on mean annual ground temperature (MAGT). 65 boreholes were selected from this dataset providing MAGT measurements within 5 years around 2010. These boreholes were then classified into three categories, i.e., boreholes with stable permafrost (MAGT is below -0.1°C), boreholes with unstable permafrost (MAGT is
between -0.1 °C and 0 °C), and boreholes with seasonal frost (MAGT is above 0 °C).

In addition, 7 boreholes collected from an existing literature (Li et al., 2016) on permafrost in the Yellow River source area, a key region in eastern QTP, are also used for the validation purpose. Ground temperatures in these boreholes were measured in the summers of 2013 and 2014, so if soil temperatures at all depths of a borehole are above 0 °C, it is considered as a borehole with seasonal frost, and otherwise one with permafrost.

In the Yangtze River source area, another key area of the QTP, recent observations of the presence/absence of permafrost in 32 boreholes acquired during the Second Tibetan Plateau Scientific Expedition and Research (Li et al., 2022) in 2020 were also used as a reference. Since permafrost on the QTP has experienced warming in recent decades (Cheng et al., 2019), some boreholes indicating the presence of seasonally frozen ground (SFG) in 2020 may be located at the permafrost zone in the 2010 map. Therefore, these 32 boreholes were not used to quantitatively validate the results, but were only used
for comparison purposes.

### 2.5 Existing QTP permafrost maps

To better evaluate the new map generated in this study, two existing permafrost distribution maps with 1 km resolution representing permafrost distribution on the QTP around 2010 were cited in this study: a new permafrost distribution map on the Tibetan Plateau (hereinafter, Zou map) simulated by the empirical temperature at the top of permafrost (TTOP) model
based on remote sensing data such as MODIS LST data from 2003 to 2012 (Zou et al., 2017), and a data-driven permafrost map (hereinafter, Wang map) (Wang et al., 2019) developed by three statistical models trained by the samples from two previous QTP permafrost distribution maps: one is the 1:4 million map of the Glaciers, Frozen Ground and Deserts in China compiled in 2006 (Wang, 2013) in which permafrost on the QTP is based on a multilinear regression model developed in 2002 (Nan et al., 2002), and the other is the above-mentioned Zou map. MODIS LST data were also used as a predictor
variable in the Wang map. Recently, the Zou map was widely used to represent permafrost distribution around 2010 and to serve as a comparison target (Hu et al., 2019; Song et al., 2020; Mu et al., 2020; Ni et al., 2021; Yin et al., 2021), and Cao et





al. (2019a) regarded the Zou map as the best performing permafrost map on the QTP in an evaluation based on an inventory of field evidence.

For the sake of simplicity, we excluded lakes and glaciers from our analysis. The spatial distribution of glaciers on the QTP comes as a subset of the Second Glacier Inventory Dataset of China (version 1.0) (2006-2011) (Guo et al., 2015), and lake data for the period 2008-2010 in Long-Term Sequence Dataset of Lake Area on the Tibetan Plateau (Zhang et al., 2017) serve as the lake boundaries in this study.

## 3 Mapping method and validation

### 3.1 The FROSTNUM/COP method and the improvements

To map the distribution of permafrost on the QTP, we applied a newly developed mapping method, namely FROSTNUM/COP (Hu et al., 2020). The general process of this method and the improvements we made in this study are described in Fig. 2. This method is based on the extended ground surface frost number (FROSTNUM) model fed by satellite temperature data (Fig. 2a,b). This method requires permafrost distribution maps for subregions, which serve as optimization objectives to obtain optimal values for a key model parameter $E$ that accounts for the heterogeneity of soils, using techniques such as spatial clustering (Fig. 2c), parametric optimization (Fig. 2d), and decision tree (Fig. 2c).

The extended FROSTNUM model determines the occurrence of permafrost by a frost number $F$:

$$F = \frac{\sqrt{DDF}}{\sqrt{DDF} + E \cdot \sqrt{DDT}} \tag{1}$$

where $DDF$ and $DDT$ represent the freezing and thawing indices (°C·day) of the ground surface temperature (GST), respectively. $E$ is a parameter accounting for a combined effect of soil property shifts from the unfrozen to the frozen state and is determined by the soil thermal properties and moisture conditions in both states. If $F$ is greater than 0.5, the ground is determined as permafrost, otherwise it is seasonally frozen ground (SFG). In the ideal case, if soil properties and conditions remain constant during the phase change, $E$ equals 1 and Eq. (1) becomes Nelson's original frost number model (Nelson and Outcalt, 1987). Although the parameter $E$ is well defined, in practice it is impossible to compute its value directly due to the lack of accurate data on soil properties and soil moisture on the QTP. Therefore, the FROSTNUM/COP method resorts to an optimization procedure to solve for $E$, which is detailed in Section 3.3. Once the driving data ($DDT$ and $DDF$) and parameters are ready, the extended FROSTNUM model is applied to map the distribution of permafrost throughout the QTP.

The original FROSRNUM/COP method uses Cohen's Kappa coefficient (Cohen, 1960) between simulation results and survey-based subregion maps as the objective function, which may lead to the equifinality problem in solving the optimal values of $E$, since the target is a binary subregion map. In this study, we specially addressed the equifinality problem. We modified the objective function to additionally include a metric that guarantees boundary consistency, and estimated the final map as a result of an ensemble of 1000 runs of parametric optimization. These processes are explained in Section 3.3.



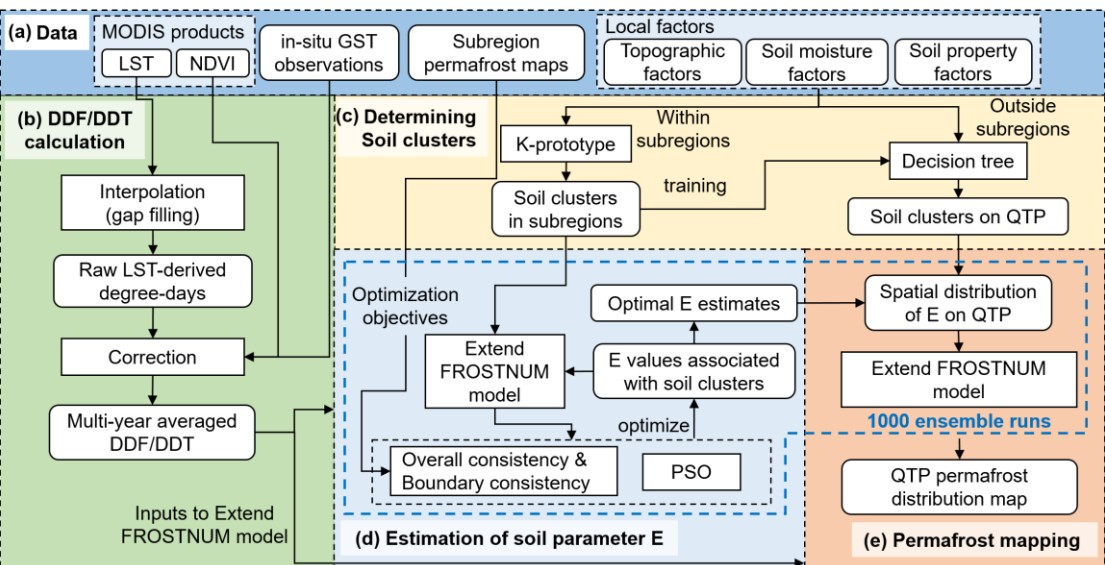

**Figure 2. Flowchart of the permafrost mapping method. (a) lists the data required in this study. (b) shows the preparation of annual ground surface freezing indices (DDF) and thawing indices (DDT) based on satellite land surface temperature (LST) data. (c) shows the process of spatial clustering of soils in subregions and prediction of the clusters in the study area based on local**
**factors. (d) shows the determination of the optimal soil parameter ($E$) values in the model using the particle swarm optimization (PSO) algorithm constrained by the given subregion maps. (e) shows the process of mapping the permafrost distribution on the QTP based on 1000 ensemble runs using the extended ground surface frost number model (FROSTNUM). Dashed blue lines enclose the improved processes over the original FROSTNUM/COP method (Hu et al., 2020), including refinement of the optimization objective and ensemble runs. The diagram has been modified from Hu et al. (2020). GST: ground surface**
**temperature; NDVI: normalized difference vegetation index.**

### 3.2 Preparation of ground surface freezing and thawing indices

The ground surface freezing indices and thawing indices were calculated based on the MOD11A1 and MYD11A1 level 3 products (version 6). Gaps in the LST products due to cloudiness have been reported to induce systematic cold biases (Westermann et al., 2012) and lead to uncertainties in permafrost mapping when using these products as inputs, so a suitable
interpolation method must be applied to fill the gaps. Despite the availability of many all-weather LST products (Zhang et al., 2021; Xu and Cheng, 2021), in this study, we chose to perform a stepwise interpolation approach based on the solar-cloud-satellite geometry (SCSG) effect (Chen et al., 2022) to interpolate the void regions in MODIS LST data. Compared with existing approaches, the SCSG-based approach is theoretically effective in tackling the case of extensive missing data (e.g. on the QTP). Moreover, this interpolation approach relies only on MODIS family data, while other all-weather LST products
rely on more data resources with various levels of uncertainty, making it advanced in uncertainty control.

In this SCSG-based stepwise interpolation approach, firstly, clear-sky LSTs are interpolated for all cloud-affected pixels (Chen et al., 2020). This clear-sky interpolation approach uses multiple temporally proximate images as reference images, with which multiple initial estimates are made for each interpolated pixel by an empirically orthogonal function method and then merged using a Bayesian approach to arrive at the best estimate of clear-sky LST equivalent. Then, a
recovering approach (Chen et al., 2022) based on the solar-cloud-satellite geometry (SCSG) effect (Wang et al., 2017; Wang





et al., 2019) is applied to impose the cloud effects on clear-sky LST equivalents. As the illumination angle of the sun and the view angle of the satellite with respect to the ground are different, some cloud-affected regions can be observed by satellite and thus LST observations in those cloudy pixels are available, thus providing important samples to consider the effects of cloudiness on the clear-sky LST equivalents. This process is realized with Multivariate Adaptive Regression Splines

(Friedman, 1991) taking as training data the pixels under cloudy conditions yet with known LST due to the SCSG effect. The predictor variables include clear-sky LST equivalent obtained in the previous step, albedo, downward shortwave radiation, net surface shortwave radiation, elevation, slope, and NDVI. As a consequence, four all-weather LST values per day are available at each 1 km by 1 km pixel. Then, a sinusoidal method (Van Doninck et al., 2011) is applied to calculate the daily mean LST based on four instantaneous LST observations and the corresponding satellite overpass times.

Due to the buffering effect of seasonal snow cover and vegetation, thermal offsets often exist between satellite-derived LST values and GST values. Therefore, an algorithm was developed to remove the influence of thermal offsets from satellite-derived LST values. In most areas of the QTP, snow cover is thin and short-lived (Wu and Zhang, 2008; Zhao et al., 2017), thus the impact of snow cover is limited, while the LST has been reported to be close to the GST during the snow-free period (Hachem et al., 2012), as also shown later in this study. Therefore, we did not take snow cover into consideration and

the DDF values were directly aggregated from the absolute value of mean daily LST below 0°C.

Nevertheless, vegetation cover is a strong buffer layer affecting the GST-LST thermal offsets, especially on the east QTP during growing seasons. Therefore, the raw LST-derived thawing degree-days, as a sum of the mean daily LST values above 0°C, cannot be used directly as the DDT as required in the extended FROSTNUM. Inspired by a study on estimating soil temperature from LST, NDVI and solar declination using linear regression models (Huang et al., 2020), we estimated

the annual DDT from raw LST-derived thawing degree-days based on a multilinear regression model where GST is a function of independent variables including LST, NDVI, and latitude at weather stations.

There are two paths for estimating the annual DDT of each pixel from the raw LST-derived thawing degree-days at that pixel. One is a form of 'one-year estimation' in which a single regression model is fitted for each year and the annual DDT is solved based on this yearly regression model. The other is a form of 'interval-based estimation', dividing a whole

year into 23 time intervals corresponding to the 16-day composite NDVI values each year. Most intervals consist of 16 days, except for the last interval. A multilinear regression is established for each time interval to estimate the thawing index for this time interval, and then the 23 thawing indices per year are summed to obtain an annual DDT.

The multilinear regression model for each time interval per MODIS grid cell has the following form trained on data at meteorological sites:

$$DDT_{i,\text{GST}}' = f(DDT_{i,\text{LST}}', N_i, L) \tag{2}$$

where $DDT_{i,\text{GST}}'$ is the ground surface thawing index for the $i$th interval of the year. $DDT_{i,\text{LST}}'$ is the thawing degree-days derived from the positive daily mean MODIS LST values at the grid cell for the $i$th interval. $N_i$ refers to the $i$th composite NDVI value in the cell. At the training stage, it is the MODIS cell in which the meteorological site is located. $L$ is the latitude. The index $i$ ranges from 1 to 23. The fitted regression functions ($f$) for all intervals are then applied to the entire



QTP to obtain interval-based DDT values for a given year, before summing them for the annual DDT for that year. To reduce the risk of single-year meteorological anomalies, annual DDF and DDT values are averaged over 2005 to 2010 to obtain multi-year averages of DDT and DDF, which are then used to drive the extended FROSTNUM model. The one-year estimation is performed on a yearly basis, rather than on an interval basis, following a similar approach of Eq. (2), without the need to sum the interval-based values.

To evaluate the effectiveness of the interval-based estimation method, we randomly divided the 131 weather stations into a training set (70%) and a testing set (30%) for 100 times. Each time we performed both interval-based estimation and one-year estimation based on the same training set, and then evaluated their prediction results using the testing set. Pearson's correlation coefficient (r), root mean squared error (RMSE), mean absolute error (MAE) and coefficient of determination (R-squared) are used as performance metrics. In addition, it should be noted that although this approach cannot completely eliminate all biases in resulting annual DDT/DDF values, residual errors can be further reduced in the optimization phase of our mapping approach, as the parameter optimization process can implicitly consider thermal offsets in the optimal values of soil parameter $E$ when targeting the survey-based subregion permafrost map.

## 3.3 Determination of optimal values of soil parameter $E$

The soil parameter ($E$) in the extended FROSTNUM model is theoretically dictated by soil moisture and thermal conditions. Soils that have similar moisture and thermal conditions are likely to possess a similar value of $E$. Therefore, the FROSTNUM/COP approach (Hu et al., 2020) applies a spatial clustering technique to group soils in subregions into several clusters based on environmental factors, and then infers $E$ for each soil cluster using the particle swarm optimization (PSO) algorithm (Wang et al., 2018) to make the simulation map best match the survey-based subregion permafrost maps. The number of clusters should be iteratively adjusted to better represent the actual distribution of soil conditions in the subregions. We predefined the number of clusters as the number of predominant soil texture types in the soil map and then would arrive at the most suitable number of clusters by evaluating the agreement of the simulated permafrost distribution, based on the number of clusters decreasing by one at a time, with subregion maps.

In this study, we followed the method developed by Hu et al. (2020) to spatially group soils from five subregions into soil clusters. As the QTP is a much larger region with more complex climate and terrain conditions than the experimental study area in the previous study (Hu et al., 2020), the environmental variables that account for the influences of local factors on permafrost distribution are slightly different in this study. Here we employed elevation, slope, NDVI, FSC, TWI, precipitation, and soil texture type, among which NDVI and FSC are newly included to further account for the influence of surface conditions that are of relatively strong heterogeneity on the QTP. Compared with Hu et al. (2020), we excluded relief degree, a topographic factor, because of its high correlation with slope. Since soil texture type is a categorical variable and the others are numerical, the k-prototype approach (Huang, 1998) was employed in clustering. In this process, lakes were excluded.





The particle swarm optimization (PSO) algorithm (Wang et al., 2018) was used to find the optimal value of $E$ associated with each soil cluster. In this population-based heuristic method, the candidate solutions are guided toward the best-known positions in the search space, thus enabling a very rapid convergence to an optimal value. In the previous study (Hu et al., 2020), Cohen's Kappa coefficient (Cohen, 1960) between the simulation map and the survey-based subregion

permafrost distribution maps was used as the only objective function. Despite the good performance achieved in the experimental study area (Gaize in Fig. 1), this relatively simple objective function will inevitably lead to equifinality in larger regions such as the QTP. In view of the fact that the Kappa coefficient well represents the overall consistency between simulation results and subregion maps, we retained Kappa coefficient ($\kappa$) and imposed a more stringent constraint on the objective function by adding a specially defined boundary consistency. The objective function is then a weighted sum of

overall consistency and boundary consistency ($\beta$):

$$F_{ob} = \omega_\kappa \cdot \kappa + \omega_\beta \cdot \beta \tag{3}$$

where $F_{ob}$ is the objective function value, $\omega_\kappa$ and $\omega_\beta$ are the weights of $\kappa$ and $\beta$ respectively, $\omega_\kappa + \omega_\beta = 1$. Since there is no straightforward way to determine $\omega_\kappa$ and $\omega_\beta$, to minimize the effects of weights, a random value between 0.2 and 0.5 was chosen for $\omega_\beta$ and 1000 parametric optimization trials were conducted. $\beta$ represents boundary consistency that measures how well the boundaries between permafrost and SFG zones in the subregion maps are represented by the simulation.

$\beta$ is defined as the numbers of 'positive boundary cells' ($N_m$) normalized by the total number of 'boundary cells' ($N_b$) identified from the survey-based subregion maps of permafrost distribution, with a range of 0 to 1:

$$\beta = \frac{N_m}{N_b} \tag{4}$$

A boundary cell is a cell on the survey-based subregion maps that satisfies two conditions in its neighborhood of size $n \times n$. First, both the types of frozen ground (permafrost and SFG) underlie its neighborhood. Second, the neighborhood contains at least two soil clusters. According to Eq. (1), the permafrost grid cells must have larger $F$ values than those of

SFG grid cells. Therefore, in the neighborhood of any 'boundary cell', the $F$ value averaged over the permafrost zone ($\overline{F_p}$) must be greater than that of the SFG zone ($\overline{F_s}$). A 'boundary cell' is 'positive' when this condition ($\overline{F_p} > \overline{F_s}$) is met, otherwise it is 'negative'.

The optimization procedure aims to maximize $\beta$ as part of the objective function, i.e., as high boundary consistency as possible of the simulated map in line with the subregion maps. In the simulated map, since prior to the optimization

procedure the DDTs and DDFs are already determined by satellite temperature data, the value of frost number $F$ of each grid cell depends on the $E$ value associated with the specific soil cluster of that cell. This means, by adjusting the $E$ values of the soil clusters within the neighborhood of a 'negative boundary cell', this cell will probably become 'positive'. In the other words, the number of 'positive boundary cells', as well as $\beta$, is a function of $E$, which makes the parametric optimization work.

To illustrate this concept, we present a simple instance of 'boundary cell' located in the survey-based map with a 3×3 neighborhood where both frozen ground types (permafrost and SFG) and two soil clusters are present (Fig. 3). The DDT-over-DDF ratios in those cells are already known as already calculated from the satellite LST data (Fig. 3). The ratios in the permafrost cells in the neighborhood are presumably higher than those on the SFG cells to resemble a scenario where permafrost persists in areas despite unfavorable climatic conditions due to favorable local factors. This cell in the center

would be considered as a 'negative boundary cell' if the $E$ values associated with the two soil clusters take 1, resulting in $\overline{F_p}$ being smaller than $\overline{F_s}$ (Fig. 3b). By adjusting $E$ values accordingly, this negative boundary cell can become positive (Fig. 3c), i.e., with a larger $\overline{F_p}$ versus $\overline{F_s}$. Thus, by enforcing such boundary consistency, an extra constraint is imposed on the resolution of the optimal $E$ values to mitigate parametric equifinality.

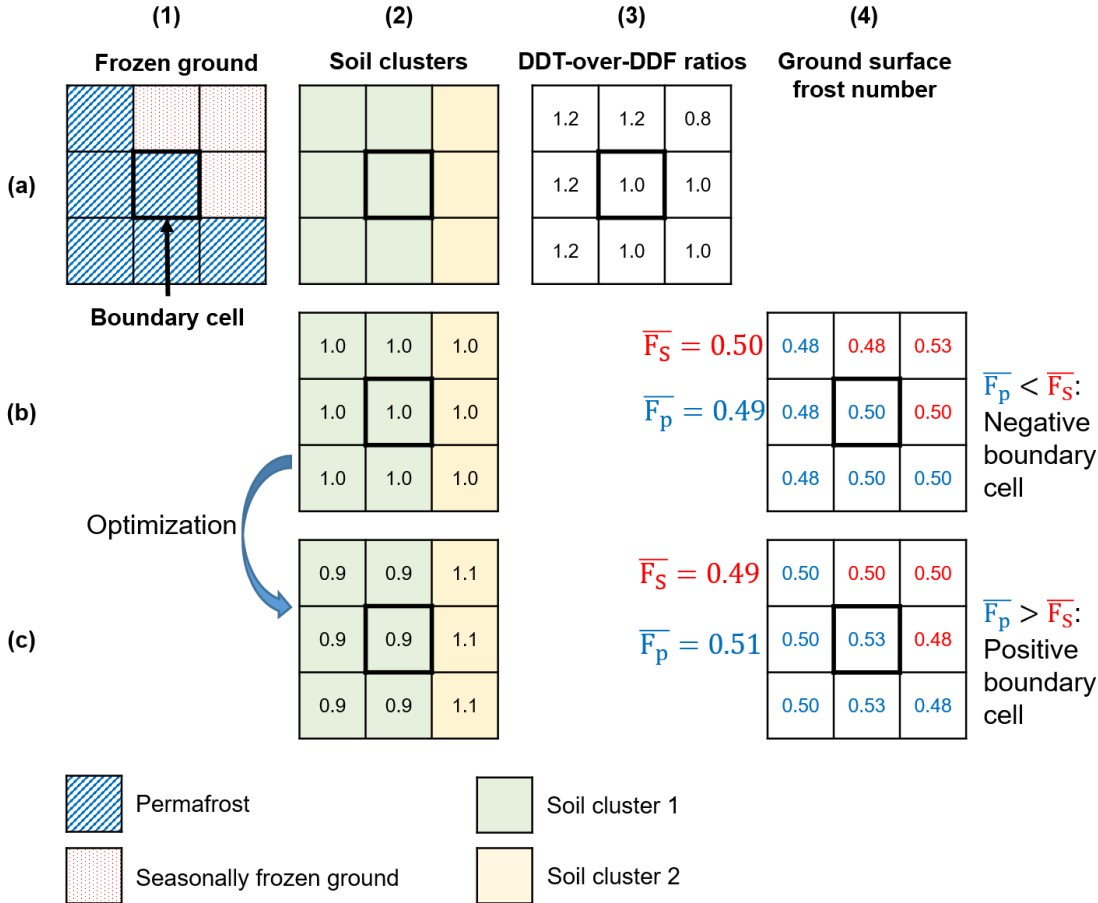

**Figure 3. Illustration explaining the concept of a boundary cell and the optimization process to improve boundary consistency. Column (1) shows a boundary cell in a survey-based map defined within a 3×3 neighborhood with permafrost and seasonally frozen ground distributed. Column (2) shows two soil clusters present in the neighborhood. The numbers on the cells indicate the values of parameter $E$ associated with the soil clusters in the cells. Column (3) shows the DDT-over-DDF ratios predetermined on the grid cells. In this case, permafrost cells have DDT-over-DDF ratios greater than one, indicating an unfavorable climate**

**condition for permafrost formation. Column (4) shows the resulting ground surface frost numbers ($F$) for the cells. $\overline{F_p}$ is an average of $F$ over permafrost cells in the neighborhood and $\overline{F_s}$ an average of $F$ over seasonal frost cells in the neighborhood. A**



**boundary cell is positive when $\overline{F_p}$ is greater than $\overline{F_s}$. Row (b) indicates a negative boundary cell when $E$ values assume 1; and row (c) shows that this boundary cell becomes positive by adjusting $E$ values. Boundary consistency improves when negative boundary cells are converted to positive cells as much as possible (Row b to row c). We add boundary consistency as part of the objective function in an effort to mitigate parametric equifinality.**

In practice, the lower and upper limits of $E$ values were set at 0.5 and 1.5, respectively. Using the improved parameter optimization approach, the optimal values of parameter $E$ can be estimated for all soil clusters occurring in the subregions. Finally, the C5.0 decision tree method was applied to predict the types of soil clusters in other regions outside the subregions on the QTP based on the same environmental factors used in spatial clustering. The soil clusters in subregions were taken as

the training set of the decision tree. Then the distribution map of soil clusters on the QTP was obtained. The distribution of soil parameter $E$ on the QTP was determined by simply looking up the optimal $E$ value associated with each soil cluster in the soil cluster distribution map on the QTP.

### 3.4 Mapping permafrost distribution and evaluation

Once the optimal $E$ values are determined for all QTP grid cells, the extended FROSTNUM model can be performed

to identify the type of frozen ground of each cell from a threshold of $F$=0.5 and then the permafrost distribution on the QTP can be mapped. However, this map can still be affected by local optima or inappropriate initial values used in the optimization. To reduce these uncertainties, the parameter optimization was carried out 1000 times and the permafrost distribution on the QTP was estimated 1000 times in response to 1000 different sets of optimal $E$ values. It should be noted that, in each of the 1000 estimations, the weight of boundary consistency in the objective function (Eq. (3)) was randomly

taken from a range of 0.2 to 0.5 since no other criteria could be used to decide an exact weight value, and the weights of overall consistency were obtained by simply subtracting the weight of boundary consistency from 1. Eventually, an ensemble permafrost map on the QTP was produced by majority voting of the 1000 estimates.

We validated the resulting map (hereafter, our map) from multiple facets. Although these survey-based subregion permafrost maps have been used as constraints in the optimization process, the optimal E values are results from all

subregion maps as a whole. Therefore, the survey-based permafrost map in each subregion is still of value for validating our map covering the entire QTP. Second, we independently validated our map using in situ permafrost presence/absence observations over a 5-year time frame around 2010. At the same time, for comparison, we also evaluated the Zou map and Wang map (Zou et al., 2017; Wang et al., 2019) with the same reference. We especially analyzed the spatial inconsistency of our map and the Zou map (Zou et al., 2017) in some typical regions. In these regions, we employed additional references

such as boreholes, satellite images, the permafrost zonation index (PZI) map (Cao et al., 2019b), and elevation distribution to further assess the performances of our map and the Zou map. Among them, satellite images can provide indicative landscape evidence on permafrost existence to some extent. Although PZI does not provide the actual permafrost extent, it can represent the probability of permafrost existence with a value ranging from 0 to 1 (Gruber, 2012). Therefore, PZI statistics can provide another means of assessing permafrost distributions. In some regions of the QTP thermally controlled by





elevation, the relationship between permafrost distribution and elevation is useful for evaluating permafrost distribution maps.

## 4 Results and discussion

### 4.1 Ground surface thawing /freezing indices

There is good consistency in terms of spatial variation, despite considerable discrepancies, between the multi-year
averaged in situ DDT values at the QTP weather stations and the corresponding annual DDT values directly aggregated from daily mean MODIS LSTs (raw LST-derived DDT) averaged over 2005-2010 prior to bias correction (Fig. 4). The discrepancies are mainly connected to the thermal offset between remotely sensed LST and GST, which has also been reported by previous studies (Luo et al., 2018; Obu et al., 2019). In contrast, the raw LST-derived DDF values align well with the in situ DDF values at QTP meteorological stations, echoing the limited impacts of thin and short-duration snow
cover on the thermal condition of permafrost on the QTP (Wu and Zhang, 2008; Zhao et al., 2017).

Table 1 shows the performances of two bias correction approaches (interval-based estimation and one-year estimation) for correcting raw LST-derived DDT based on 100 times of randomly split testing sets. By evaluating against annual in situ DDT values at QTP sites, the annual DDT values obtained by the interval-based approach has generally lower errors (mean RMSE: 437 °C·day, mean MAE: 349 °C·day) and better linear correlation (mean r: 0.94) than those by the one-year
estimation approach (mean RMSE:486 °C·day, mean MAE: 438 °C·day, mean r: 0.92). The ranges of metric values for the interval-based estimation are all narrower than those for the one-year estimation, indicating consistent improvements in performance at all sites. This demonstrates the advantage of adopting the interval-based estimation over the one-year estimation approach in reducing biases in MODIS LST-derived DDT data.

Obvious negative biases can be found in the raw LST-derived annual DDT (Fig. 5a) which generally underestimates
the in situ annual DDT at most of the QTP weather stations from 2005 to 2010. However, after correction using the interval-based estimation approach, the negative biases have been well eliminated and the corrected data points are concentrated along the 1:1 line as shown in Fig. 5b with an increased R-squared value of 0.89 from 0.74 before the correction. More importantly, the MAE value (334 °C·day) of the corrected DDTs is about one-third of the value (889 °C·day) measured before the correction, i.e., dropping from 23.3% in relative error to 8.8%, below the 10% level. In addition, raw DDT data
points with large deviations leading to a great value of 1072 °C·day in RMSE have been effectively corrected arriving at a RMSE of 421 °C·day after correction. Most corrected data points fall into the ±400 °C·day intervals (about 10%) around the 1:1 line, indicating a well-controlled level of error.

Figure 6 depicts the spatial patterns of the estimated multi-year averages of annual DDT, annual DDF and the DDT-over-DDF ratio over the QTP at a 1km resolution. The distributions of annual DDT and DDF are in close agreement with
elevation (Fig. 1), which is one of the main factors controlling ground temperature distribution over the QTP. In general, annual DDT decreases and annual DDF increases with rising elevation. Over the relatively flat high plain, lying 80-90°E and





33-37°N, the annual DDT shows moderate latitudinal zonality, declining as latitude increases, whereas the annual DDF shows the opposite. This is an indication of the influence of solar radiation on ground temperature. The vast area and complex topography of the QTP result in a wide spectrum of annual DDT (up to 9000 °C·day) and DDF (up to 8000 °C·day).

Most regions of the QTP lie between 3000 and 5500 m a.s.l., where DDT and DDF values are mostly distributed between 1000 °C·day and 2500 °C·day. High DDT values appear in the low mountains in the southeast QTP, in the Qaidam Basin in the north, and in the south valleys, while high DDF values appear on the Qiangtang Plateau in the northern QTP and in the high mountain areas, favoring the formation of permafrost. The ratios of DDT-over-DDF indicate climatic controls on permafrost preservation (Fig. 6c). If there were no local factors influencing permafrost distribution, regions with a DDT-

over-DDF ratio <1 would be regarded as where permafrost potentially forms.

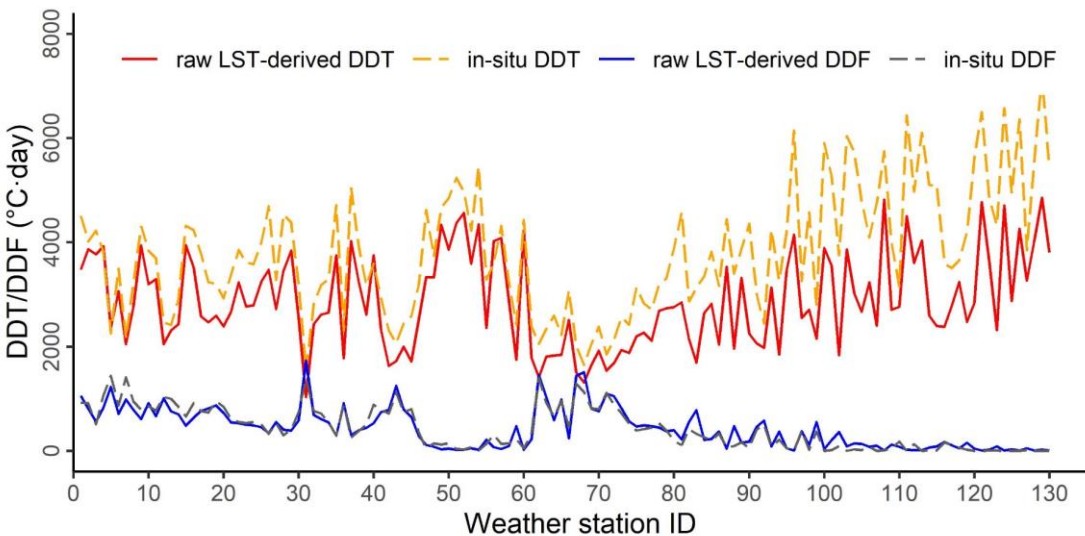

**Figure 4. Observed discrepancies between annual thawing degree-days aggregated from interpolated MODIS LST data (raw LST-derived DDT) and those from in situ observations of ground surface temperature (in situ DDT) at the QTP weather stations, whereas raw LST-derived annual freezing degree-days (raw LST-derived DDF) closely approximate the in situ annual DDF. The**
**ordinate indicates the annual thawing/freezing degree-days averaged over the period 2005 to 2010; and the abscissa shows 131 weather stations available on the QTP.**

**Table 1. Comparison of performance between two bias correction approaches (interval-based and one-year) for the raw LST-derived thawing index sequences based on 100 trials with a random split of the training and test data sets. The values indicate the**
**metric means from the 100 random trials, and the values in parentheses represent ranges. The training sets consist of 70% of the available stations, and the metric values provided were calculated over the remaining 30% as testing stations. r: Pearson's correlation coefficient; RMSE: root mean square error; MAE: mean absolute error.**

|  | r | RMSE (°C·day) | MAE (°C·day) |
|---|---|---|---|
| Interval-based estimation | 0.94 (0.88~0.97) | 437 (344~554) | 349 (252~458) |
| One-year estimation | 0.92 (0.85~0.97) | 486 (309~671) | 368 (240~509) |

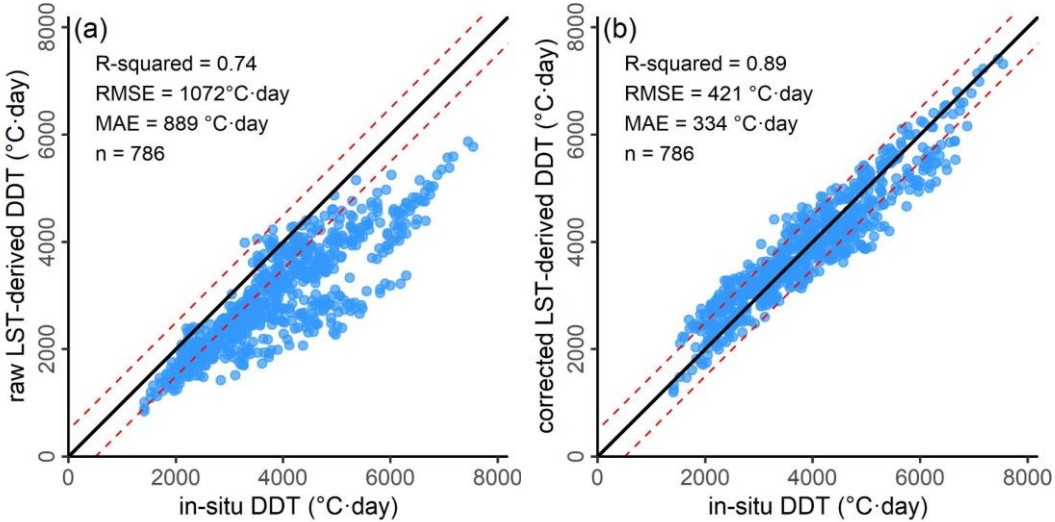

**Figure 5. Improved agreement in MODIS-LST-derived annual DDT values after bias correction with annual in situ DDT values at 131 QTP stations. (a) Before bias correction; (b) after applying the interval-based correction approach. Data points represent annual DDT values over 2005-2010 from the 131 stations. The red dashed lines outline a range of ~10% (±400 °C·day) from the diagonal line (black solid line).**

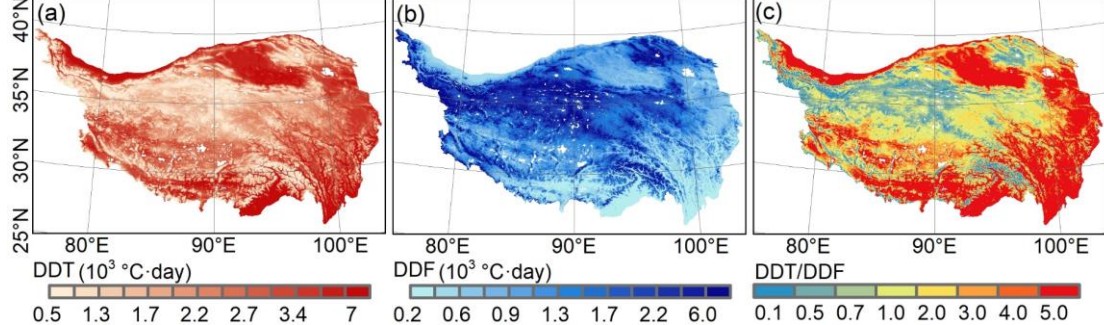

**Figure 6. Spatial distributions of (a) annual DDT, (b) annual DDF, and (c) the ratio of DDT-over-DDF, averaged from 2005 to 2010 on the QTP. Lakes were excluded and are shown blank, while glaciers were included. Regions with a DDT-over-DDF ratio of <1 are climatically favorable for permafrost formation.**

### 4.2 Soil clusters

440        Eight soil clusters were determined by the k-prototype approach in the five subregions (Fig. 7), where lakes were excluded. The dominant soil clusters in each subregion differ from each other (Table 2). Clusters 3 and 1 are dominant in West Kunlun, clusters 2 and 7 in Gaize, clusters 7 and 1 in Aerjin, clusters 8 and 7 in G108, and cluster 8 in Wenquan. This implies the distinctions of climatic and geographic conditions in these subregions.

        Soils in one cluster share more similar characteristics to each other, as reflected by a single value of model parameter *E,*
than those in other clusters. Figure 8 presents the primary characteristics of the soil clusters in the five subregions. Clusters 1, 2, and 3 differ in slope and TWI, but are all characterized by relatively high elevation (about 5000 m), low vegetation cover



(NDVI < 0.2), thin snow cover (FSC < 10), and aridity (precipitation < 200mm), which generally may represent high plateaus. Cluster 4, with the highest TWI, represents the valley with low elevation and moderate slope. Cluster 5 has the highest elevation, highest FSC, and lowest NDVI (even below zero), thus it may represent high mountains covered by thick snow cover or glaciers. Cluster 6 has very varied elevations and steep slopes, and is on the hillslopes of high mountains. Except for the much lower TWI, cluster 7 is similar to cluster 4, and from the soil cluster map (Fig. 7), we can find that cluster 7 is often distributed around cluster 4 representing valleys. Therefore, it is likely that cluster 7 represents gentle slopes adjacent to valleys. Cluster 8, mainly distributed in the two subregions (G308 and Wenquan) on the east QTP and characterized by the highest NDVI, highest precipitation, and lowest elevation, represents the regions with better hydrological, thermal and vegetation conditions on the east QTP.

The distribution of soil clusters throughout the QTP (Fig. 7) was predicted by the decision tree method. As summarized in Table 2, soil cluster 8 covers the largest area of about 37.76% of the QTP and is mainly distributed in the east QTP, which is related to the training samples of this cluster that are mainly located in G308 and Wenquan on the eastern QTP. Soil clusters 2, 3, 6, and 7 cover roughly the same area, about 10% of the whole QTP, followed by clusters 1 (6.94%) and 4 (6.20%). Soil cluster 5, representing glaciers and regions covered by thick snow cover, occupies the least area (2.85%), which is in line with a previous study according to which thick snow covered only represents a rather small part of the QTP (Dai et al., 2018).

The optimal soil parameter $E$ values associated with all soil clusters (Table 3) were obtained from the ensemble runs. The ranges of the optimal values are relatively narrow for all soil clusters, suggesting that equifinality is well mitigated due to a well-constrained objective. The mean values as the optima for clusters 4 and 5 are greater than 1, with an implication of unfavorable local conditions for permafrost formation or preservation. For example, thermal erosion occurring near rivers in cluster 4, which represents valleys, and the insulation effect of snow cover in cluster 5 are not beneficial for permafrost formation and preservation. The $E$ values of clusters 1, 3, and 8 are relatively lower, which highlights the existence of favorable local environments for permafrost formation in these regions. Some local factors, such as high elevation in clusters 1 and 3, and high precipitation in cluster 8, are beneficial for permafrost preservation (Zhang et al., 2021), the same as interpreted by their lower $E$ values.

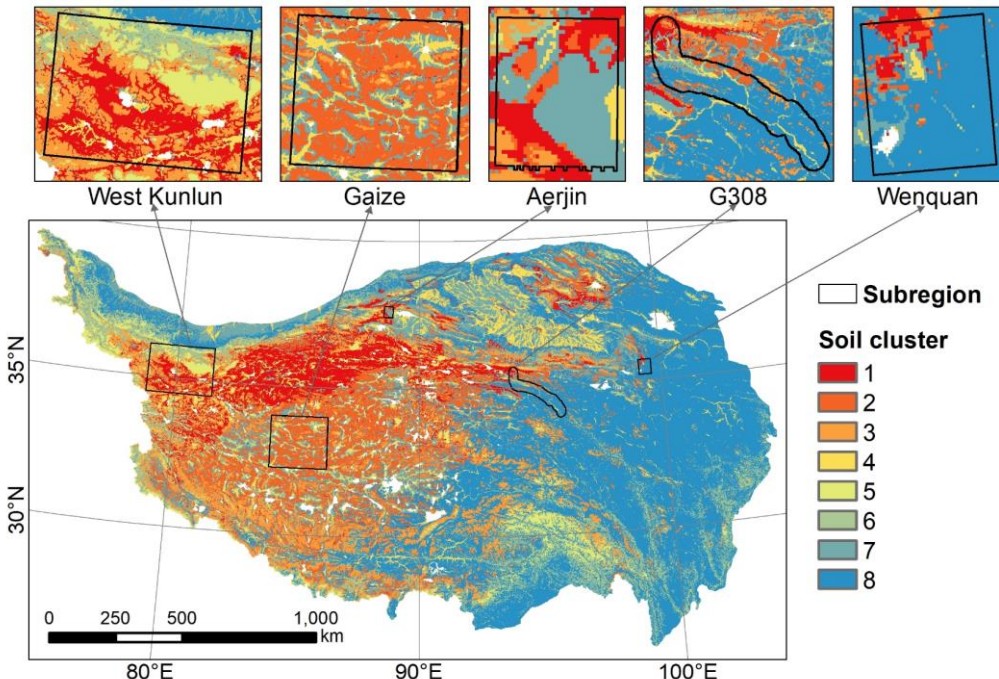

**Figure 7. Resulting soil clusters in the five subregions and the predicted distribution of clusters throughout the QTP. A total of eight clusters were determined. Each soil cluster represents unique traits as reflected by a distinct value of model parameter $E$.**


**Table 2. Area percentages of soil clusters in each subregion, over all subregions, and over the entire QTP.**

| Region | Cluster 1 | Cluster 2 | Cluster 3 | Cluster 4 | Cluster 5 | Cluster 6 | Cluster 7 | Cluster 8 |
|---|---|---|---|---|---|---|---|---|
| West Kunlun | 29.53 | 2.02 | 30.65 | 4.21 | 16.21 | 14.13 | 1.89 | 1.37 |
| Gaize | 1.11 | 57.75 | 5.34 | 12.62 | 0.12 | 0.44 | 22.55 | 0.06 |
| Aerjin | 22.97 | 12.39 | 8.51 | 3.88 | 0.40 | 2.88 | 48.76 | 0.20 |
| G308 | 6.24 | 9.56 | 4.40 | 7.29 | 0.00 | 0.12 | 17.24 | 55.15 |
| Wenquan | 7.27 | 6.37 | 0.72 | 1.62 | 0.00 | 0.04 | 9.01 | 74.97 |
| All subregions | 14.63 | 25.93 | 16.27 | 7.93 | 7.17 | 6.44 | 12.91 | 8.72 |
| QTP | 6.94 | 11.45 | 11.69 | 6.20 | 2.85 | 9.31 | 13.79 | 37.76 |

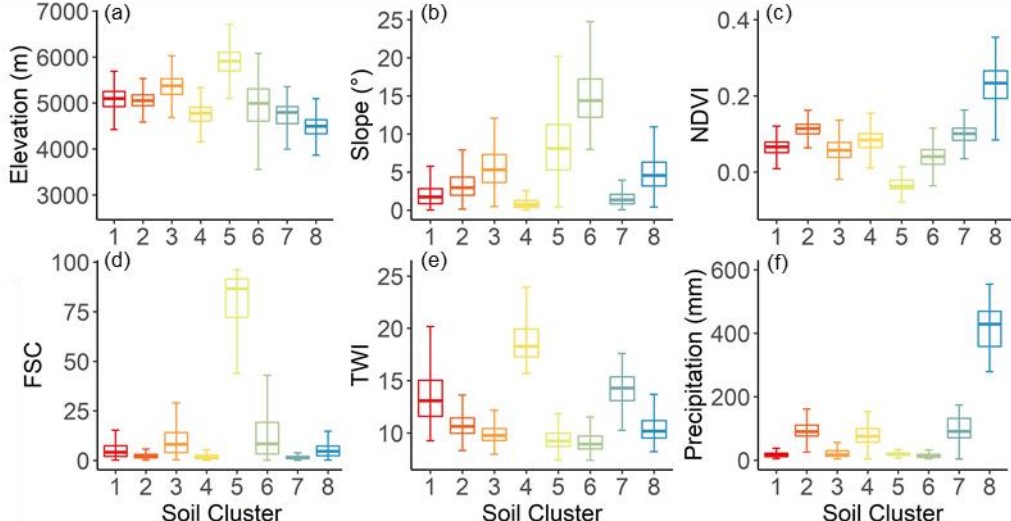

**Figure 8. Characteristics of the soil clusters determined with respect to the factors involved: (a) elevation; (b) slope; (c) normalized difference vegetation index (NDVI); (d) fractional snow cover (FSC); (e) topographic wetness index (TWI); (f) precipitation. All**
**clusters are shown in different colors in correspondence with those in Fig. 7. The center line in the box shows the median, the box shows the lower and upper quartiles, and the whiskers extend to the minimum and maximum data values.**

**Table 3. The ranges and mean values as the most optimal values of the soil parameter *E* associated with the 8 soil clusters resulted**
**from 1000 optimization trials.**

|  | Cluster 1 | Cluster 2 | Cluster 3 | Cluster 4 | Cluster 5 | Cluster 6 | Cluster 7 | Cluster 8 |
|---|---|---|---|---|---|---|---|---|
| Max | 0.778 | 0.831 | 0.784 | 1.014 | 1.149 | 1.011 | 0.938 | 0.785 |
| Min | 0.757 | 0.820 | 0.745 | 1.000 | 1.061 | 0.931 | 0.932 | 0.778 |
| Mean | 0.768 | 0.822 | 0.748 | 1.007 | 1.073 | 0.946 | 0.938 | 0.779 |

## 4.3 The resulting 2010 permafrost distribution map on the QTP

Figure 9 shows the resulting permafrost distribution map on the QTP at a spatial resolution of 1km, where permafrost underlies about $1.086 \times 10^6$ km$^2$, about 41.17% of the QTP, while SFG occupies about $1.447 \times 10^6$ km$^2$, accounting for 54.85%
of the total QTP area. The non-frozen ground is about $2.24 \times 10^4$ km$^2$, and the rest is glaciers about $4.08 \times 10^4$ km$^2$ and lakes about $4.17 \times 10^4$ km$^2$.

The map shows that permafrost is prevalent throughout the north-central QTP, especially on the Qiangtang Plateau. In the north, the Qaidam Basin is occupied by SFG due to the low altitude and breaks the permafrost continuity extending northward to the Qilian Mountains. From the central Qiangtang Plateau southward, the spatial continuity of permafrost tends
to decline due to the decrease in both latitude and altitude. Near the permafrost zone in the Bayan Har Mountains and Tanggula Mountains in the eastern QTP, SFG occurs extensively in the river source areas, namely the Three-River





Headwaters Region, probably as a result of low latitude and unfavorable local factors such as thermal erosion accompanying water flows. Given that the DDT-over-DDF ratios in these regions are often greater than 1 (Fig. 6c), it can be inferred that permafrost in the river source areas (e.g. the Yangtze River source area) may be thermally vulnerable and prone to

degradation under climate warming (Zhang et al., 2022). On the southern QTP, permafrost is sporadically distributed at high elevations, mainly in the high mountains of the Eastern Himalayan Syntaxis, Gangdise Mountains, and Himalaya Mountains. Only a small amount of non-frozen ground exists in the southeastern periphery of the QTP.

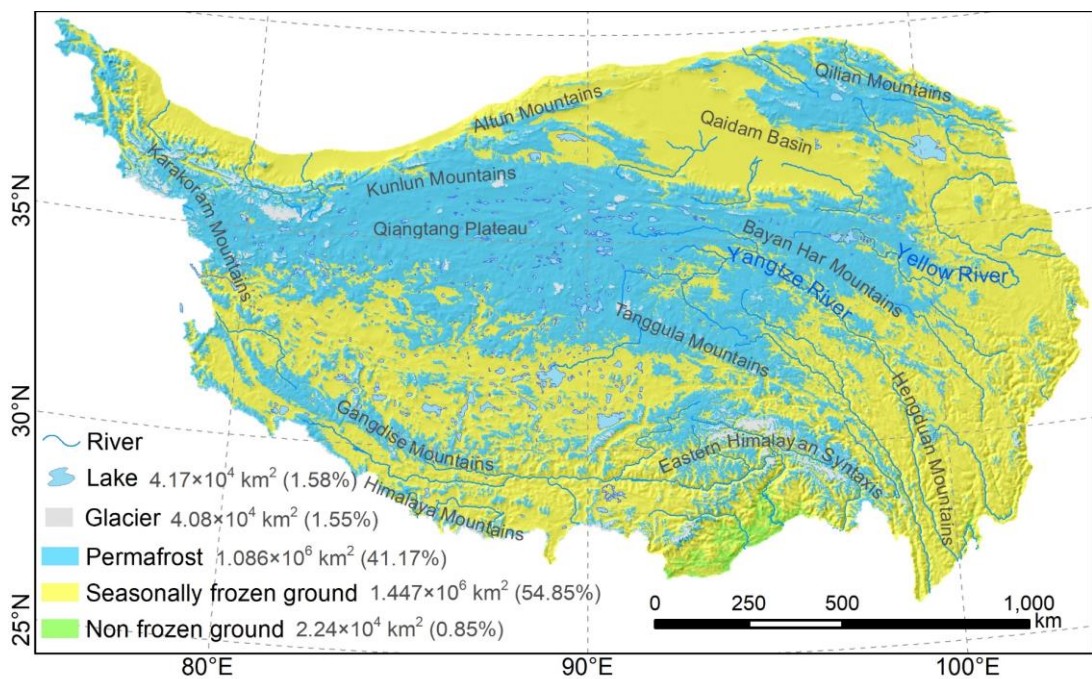

**Figure 9. Map of permafrost distribution over the QTP in 2010 (our map) produced in this study. Areas as well as percentages follow the legends. The map displays hill-shading with elevation.**

### 4.4 Assessment based on survey maps and borehole data

Our map shows substantial spatial agreement with the survey-based permafrost maps in all subregions (Fig. 10) with a

high Cohen's Kappa coefficient ($\kappa$) of about 0.74 (Table 4.), significantly exceeding that of the Zou map (Zou et al., 2017) and the Wang map (Wang et al., 2019) whose Kappa values are 0.55 and 0.50, respectively. Overall, compared with the survey-based subregion permafrost maps (Fig. 10a) and our map (Fig. 10b), the Zou map obviously overestimates the permafrost extents in Gaize and Aerjin and underestimates the permafrost extent in G308 (Fig. 10c), while severe overestimation in permafrost extent occurs in all five subregions in the Wang map (Fig. 10d).



More specifically, in permafrost-dominated West Kunlun, our map and the Zou map slightly overestimate the extent of SFG around the lake, while the Wang map slightly underestimates the extent of SFG. There are only small differences between the three maps in West Kunlun, with almost the same $\kappa$ for all three maps (0.62 for our map, 0.63 for both the Zou map and Wang map). All three maps are based on satellite LST products, which in general can capture the patterns of surface ground temperature. Although different interpolation methods were applied to fill in the gaps existing in the satellite

LST data, and consequently affected the accuracies of the driving data for the mapping methods to a varying degree, the cold climate of West Kunlun makes the effects of the biases in the input data on final permafrost distribution negligible, as most areas there have LST values well below zero.

      In Gaize, our map shows superior performance ($\kappa = 0.71$) to that of the Zou map ($\kappa = 0.48$) and Wang map ($\kappa = 0.43$), as both the Zou and Wang maps severely overestimate permafrost distribution, whereas our map matches well with the

survey-based map in this region (Fig. 10b). The same situation occurs in Aerjin, and the comparison with the survey-based map indicates a much lower $\kappa$ for the Zou map (0.38) and Wang map (0.00) than our map (0.71). Gaize has a warmer climate than West Kunlun and contains the southern limit of continuous permafrost. Therefore, the influence of input data quality and local factors on permafrost preservation is more profound in this area. The overestimated permafrost extents in Gaize and Aerjin in both the Zou map and Wang map may most likely be related to the relatively lower quality of the

interpolated LST data as model drivers and insufficient consideration of local factors. Compared with the harmonic analysis of time series (HANTS) algorithm (Xu et al., 2013) adopted in the Zou and Wang maps to reconstruct the missing LST data under clear sky assumptions, the SCSG-based interpolation method with full consideration of cloud effects on LST used in this study has been shown to be sufficiently accurate and effective in handling large areas of missing data (Chen et al., 2022). In addition, the daily GST data required in the Zou map and Wang map are the weighted sum of four MODIS LST

observations per day through an empirical linear formula based on in situ observations from only three automatic weather stations in the central QTP (Zou et al., 2014). Unlike the method in this study used to estimate GST data that relies on 131 weather stations over the QTP and comprehensively accounts for influential factors such as NDVI and latitude, this relatively simple treatment of GST estimates in the Zou map and Wang map may lead to considerable systematic biases in some regions and ultimately result in large uncertainties in the final permafrost distribution, especially in warm permafrost

regions that are highly susceptible to thermal perturbations (e.g. the Gaize subregion). Furthermore, in the case of the Wang map using statistical learning methods, uncertainties also arise from training samples selected from two previous QTP permafrost distribution maps compiled in different years being more than a decade apart and subject to varying levels of uncertainty (Ran et al., 2012; Zou et al., 2017). All these factors combined compound the overestimation of permafrost in Gaize in the Wang map.

In G308, the Wang map ($\kappa = 0.68$) indicates more permafrost areas than the local survey map, while both the Zou map ($\kappa = 0.48$) and our map ($\kappa = 0.68$) show fewer permafrost areas, being more evident in the Zou map. Soil thermal regimes in G308 are strongly affected by rivers and vegetation cover, and the effects have been well accounted for in our mapping approach, but are missed in the Zou map. In Wenquan, both our map ($\kappa = 0.70$) and the Zou map ($\kappa = 0.65$) perform



generally satisfactorily, with a slight overestimation of permafrost extent. In contrast, the overestimation is more server in
the Wang map ($\kappa = 0.46$), which may also be due to misrepresentation of training data it used.

The maps were also verified by 72 permafrost presence/absence observations obtained by boreholes drilled within a 5-year time frame around 2010 (Li et al., 2016; Zhao et al., 2021) and the resulting measures are listed in Table 5. Only our map shows good agreement ($\kappa = 0.43$) with the borehole observations in terms of $\kappa$, while the Zou map ($\kappa = 0.30$) and Wang map ($\kappa = 0.14$) have unsatisfactory performances. According to the borehole observations, SFG is underrepresented in all
three maps, among which our map achieves the best performance with 54.5% accuracy in predicting SFG locations, in contrast to the worst-performing Wang map, in which only 1 out of 11 seasonal frost boreholes is correctly identified. Contrary to this, permafrost locations are overestimated in three maps, indicated by the relatively high false positive (permafrost) rates (45.5% for our map, 54.5% for the Zou map, and 90.9% for the Wang map). As permafrost underlies most of the borehole locations used for validation (Fig. 10), the Wang map predicts almost all locations as permafrost with no
discretion as indicated by a 100% true position rate (Table 5), and consequently leads to an inflated accuracy of 86.1%, which is the highest among the three maps.

In our map, 2 out of 6 false negatives (wrongly identified as SFG) are the boreholes with unstable permafrost located in the SFG zone close to the permafrost boundary. But in the Zou map, 8 false negatives are all boreholes with stable permafrost, and those with unstable permafrost are located in the permafrost zone. If we excluded all 4 boreholes with
unstable permafrost from the evaluation, the false negative rate of our map would drop from 9.8% to 7% and $\kappa$ would rise from 0.43 to 0.49. However, the false negative rate and $\kappa$ of the Zou map would remain almost unchanged, leaving an even higher false negative rate (~13%) than that of our map (~7%). This verification against borehole observations may be biased by the scale mismatch between a site and a 1 km $\times$1 km grid cell. Nevertheless, those evidences as a whole suggest a decent performance of our map in predicting the type of frozen ground than the two counterpart maps.






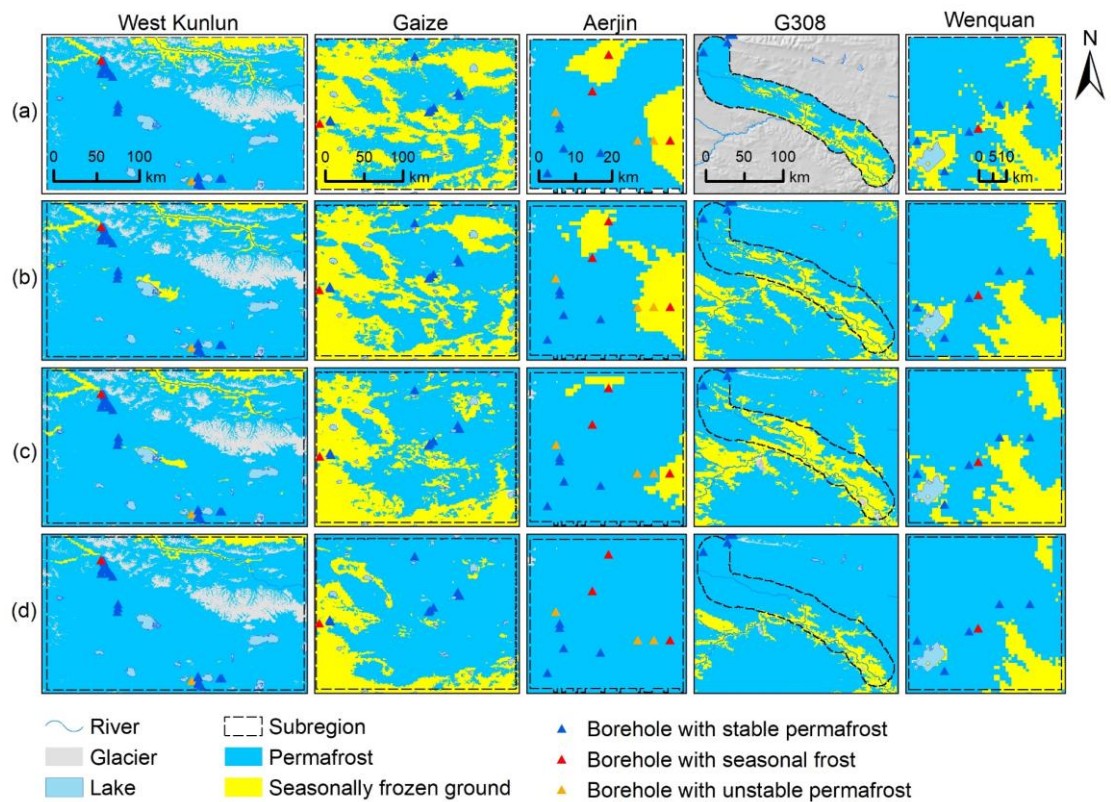

**Figure 10. Spatial distributions of frozen ground in subregions from (a) survey-based maps, (b) our map, (c) the map from Zou et al. (2017), and (d) the map from Wang et al. (2019). Triangle symbols mark the locations of boreholes drilled around 2010 where the type of frozen ground was identified.**


**Table 4. Kappa values measured between the evaluated maps (our map, Zou map, and Wang map) and survey-based permafrost distribution maps in the subregions.**

|  | West Kunlun | Gaize | Aerjin | G308 | Wenquan | All subregions |
|---|---|---|---|---|---|---|
| Our map | 0.62 | 0.71 | 0.71 | 0.68 | 0.70 | 0.74 |
| Zou map | 0.63 | 0.48 | 0.38 | 0.46 | 0.65 | 0.55 |
| Wang map | 0.63 | 0.38 | 0.00 | 0.68 | 0.46 | 0.50 |

**Table 5. Measures of confusion matrices describing the performance of the evaluated maps (our map, Zou map, and Wang map) at the borehole locations. To fit the binary classification, permafrost is regarded as positive and seasonally frozen ground negative. n=72.**


|  | Our map | Zou map | Wang map |
|---|---|---|---|
| True positives | 55 | 53 | 61 |



| False positives | 5 | 6 | 10 |
|---|---|---|---|
| True negatives | 6 | 5 | 1 |
| False negatives | 6 | 8 | 0 |
| True positive rate | 90.2% | 86.9% | 100.0% |
| False positive rate | 45.5% | 54.5% | 90.9% |
| True negative rate | 54.5% | 45.5% | 9.1% |
| False negative rate | 9.8% | 13.1% | 0.0% |
| Accuracy | 84.7% | 80.6% | 86.1% |
| Cohen's Kappa | 0.43 | 0.30 | 0.14 |

## 4.5 Cross-comparison with the Zou map

In light of the widely recognized performance of the Zou map, which has been used as a reference of the present QTP
permafrost distribution in many studies (Hu et al., 2019; Song et al., 2020; Mu et al., 2020; Ni et al., 2021; Yin et al., 2021),
we further compare our map to the Zou map (Zou et al., 2017). The permafrost distributions in the two maps are generally
consistent, although there are discrepancies in some regions (Fig. 11) that are mainly in the transition region between the
continuous permafrost zone over the Qiangtang Plateau to the north and the SFG zone to the south. In addition, the
headwaters regions of China's major rivers (regions c, d in Fig. 11) on the eastern QTP show remarkable spatial
inconsistency between the two maps. Those headwater regions have been reported to be the critical regions where permafrost
is more vulnerable and very sensitive to climate change (Jin et al., 2011; Zhang et al., 2021).

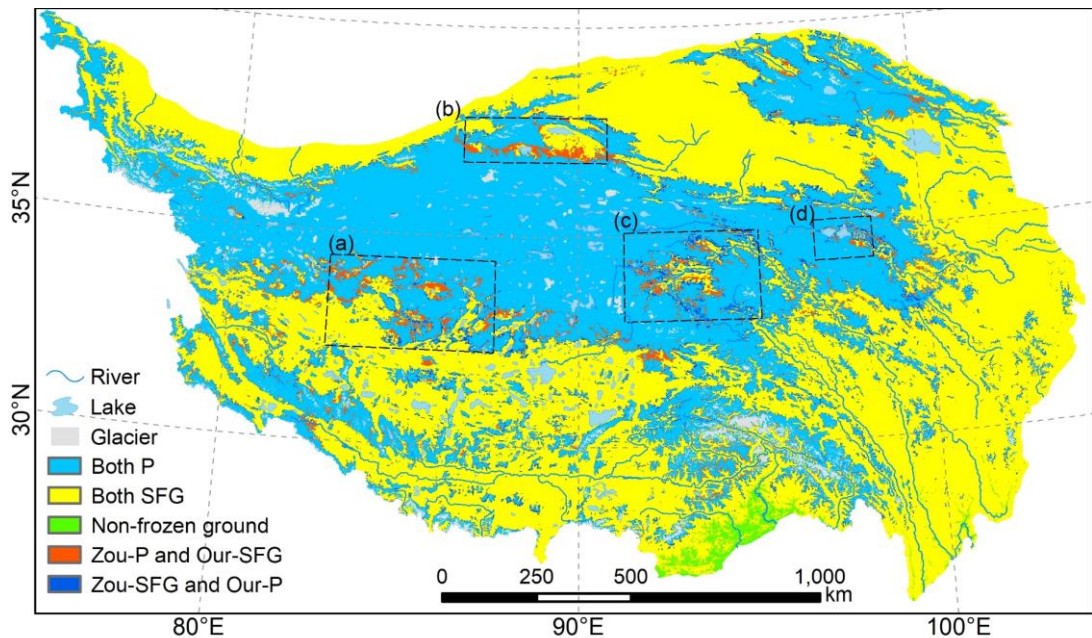

**Figure 11. Inconsistencies in the spatial distribution of frozen ground types between our map and the Zou map. "Both P" represents areas identified as underlying permafrost in both maps; "Both SFG" represents areas identified as seasonally frozen ground (SFG) in both maps; "Zou-P and Our-SFG" represents areas identified as underlying permafrost in the Zou map but SFG in our map; "Zou-SFG and Our-P" represents SFG identified as SFG in the Zou map but underlying permafrost in our map. The dashed boxes indicate areas of tremendous inconsistency. (a) Gaize and its vicinity, (b) the areas between the Altun Mountains and the Kunlun Mountains, (c) the headwaters of the Yangtze River, and (d) the headwaters of the Yellow River.**

In and around Gaize (region a in Fig. 11), a larger extent of permafrost is simulated in the Zou map than in our map. Comparisons of the two maps with the survey-based Gaize map (Fig. 10) have already confirmed a much better performance of our map in this region than the Zou map, due to the use of the survey-based Gaize map as part of the constraints on the mapping approach of our map (Table 4). The vicinity of Gaize is also characterized by a relatively flat plateau with an arid climate and low vegetation cover, very similar to the Gaize subregion. It can be deduced that, in and around the Gaize subregion, our map could probably show better accuracy than the Zou map.

In the areas between Altun and Kunlun Mountains (region b in Fig.11, Fig. 12) containing the Aerjin subregion, it seems that much more SFG is estimated in our map than the Zou map. Compared with the survey-based Aerjin subregion map (Fig. 10), the Zou map obviously underestimates the extent of SFG, and our map shows a better performance despite a slight overestimation of SFG extent. According to borehole measurements (Fig. 10) in the Aerjin subregion, some locations have a MAGT of around -0.1 to 0 °C, and a borehole location has even revealed a MAGT above 0 °C, but it falls within a permafrost zone in the survey-based subregion map. Despite this imprecision, it at least follows that permafrost in this large region is extremely thermally unstable. We especially inspected the distinct zones identified as underlying permafrost in the Zou map but SFG in our map (Fig. 12a), where the DDT-over-DDF ratios are around 1.3 (Fig. 12b) and clusters 4 and 7 corresponding with $E$ values of about 1.07 and 0.94, respectively, predominate (Fig. 7). Those characteristics are very similar to the SFG zone in the survey-based Aerjin map. Following Eq. (1), surface frost numbers are calculated to be less



615    than 0.5 in these distinct areas, which is suggestive of no permafrost presence. From the satellite image (Fig. 12c), it can be seen that rivers are well developed in the basins, thus providing clues to permafrost degradation in those areas. Overall, based on limited evidence, our map shows more acceptable distribution characteristics in this region than the Zou map. However, more field studies are necessary to provide more direct evidence to strengthen our understanding toward permafrost distribution in this critical region.

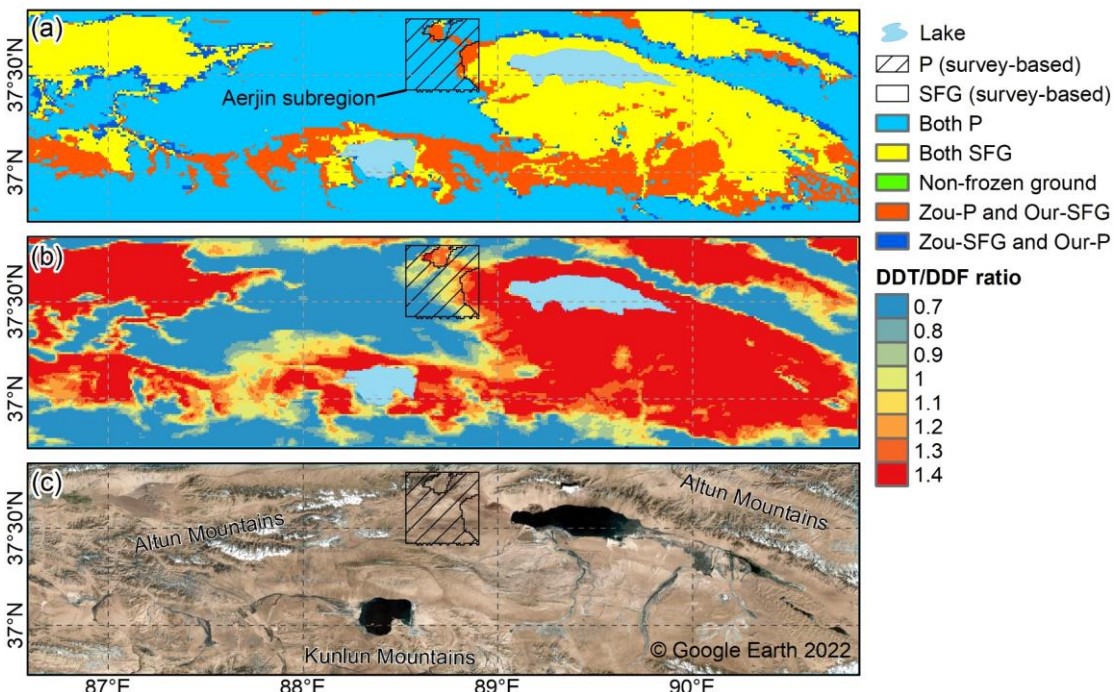

**Figure 12. Elaborate maps of the areas between Altun Mountains and Kunlun Mountains (region b in Fig. 11) showing (a) detailed spatial differences in permafrost distribution between our map and the Zou map, (b) the ratios of DDT-over-DDF, and (c) a satellite image covering this region from Google Earth. The box indicates the Aerjin survey area with the availability of a survey-based permafrost map. P (survey-based) and SFG (survey-based) represent permafrost/seasonally frozen ground, respectively, on the survey-based Aerjin subregion map. For Both P, Both SFG, Zou-P and Our-SFG, and Zou-SFG and Our-P, the same notations apply as in Fig. 11.**

In the source areas of the Yangtze River (region c in Fig. 11, Fig. 13), the riparian zones are generally identified as SFG in both our map and the Zou map. However, the SFG zones in our map spread on both sides along the rivers, while they are only distributed on one side of the rivers in the Zou map. In this region, 33 permafrost presence/absence observations have been collected. Among them, 32 were drilled in 2020 during the Second Tibetan Plateau Scientific Expedition and Research campaign (Li et al., 2022), so they have not been used in the above quantitative validations. The only remaining one comes from the dataset of Zhao et al. (2021) (black circle in Fig. 13) representing the 2010 state and is correctly identified for the frozen soil type in both maps. As for the 32 boreholes drilled in 2020, 2 of 6 boreholes with seasonal frost and 24 of 26 boreholes with permafrost are consistently identified in our map, whereas in the Zou map, no borehole with seasonal frost and 24 boreholes with permafrost are correctly identified. The misidentified boreholes are located near the





boundary of the permafrost zone on our map, whereas on the Zou map they are mostly located within permafrost zones. We also noted that two upstream boreholes at 4870 m (Li et al., 2022) located within a permafrost zone in both maps (red box in Fig. 13) revealed seasonal frost in 2020. Considering the potential impact of climate warming occurring in this region over the past decade, it is possible that permafrost degradation has occurred at the two upstream borehole locations, leading to a
conversion of permafrost in 2010 to SFG in 2020. In these areas, the occurrence of permafrost degradation usually recedes to upstream areas. In other words, at reasonable inference, in 2010 permafrost remained in upstream areas and SFG had occurred in downstream areas along the rivers, as has been well depicted in our map (Fig. 13a).

We further examined the two maps in the Yangtze River source areas using a PZI approach (Cao et al., 2019b). A higher PZI indicates a high possibility of permafrost presence (Gruber, 2012). The PZI map used here was compiled based
on 1475 in situ observations (Cao et al., 2019b), and many in situ observations were obtained between 2005-2018 near the Nation Highway G109 traversing the source areas of the Yangtze River, which makes the PZI map a possible reference in this region. We analyzed the distributions of PZI values in the permafrost and SFG zones of this region in our map and the Zou map (Fig. 13d). It is found that the PZI distribution for permafrost in our map is close to that in the Zou map. However, for the PZI values in SFG regions, the lower and upper quartiles in the Zou map are 0.36 and 0.66, respectively, while those
in our map are 0.34 and 0.53. The upper quartile in SFG regions (0.66) in the Zou map even overlaps the lower quartile in permafrost regions being around 0.55, which is likely statistically unreasonable. By contrast, the PZI statistics for both frozen ground types in our map turn out to be more distinguishable.

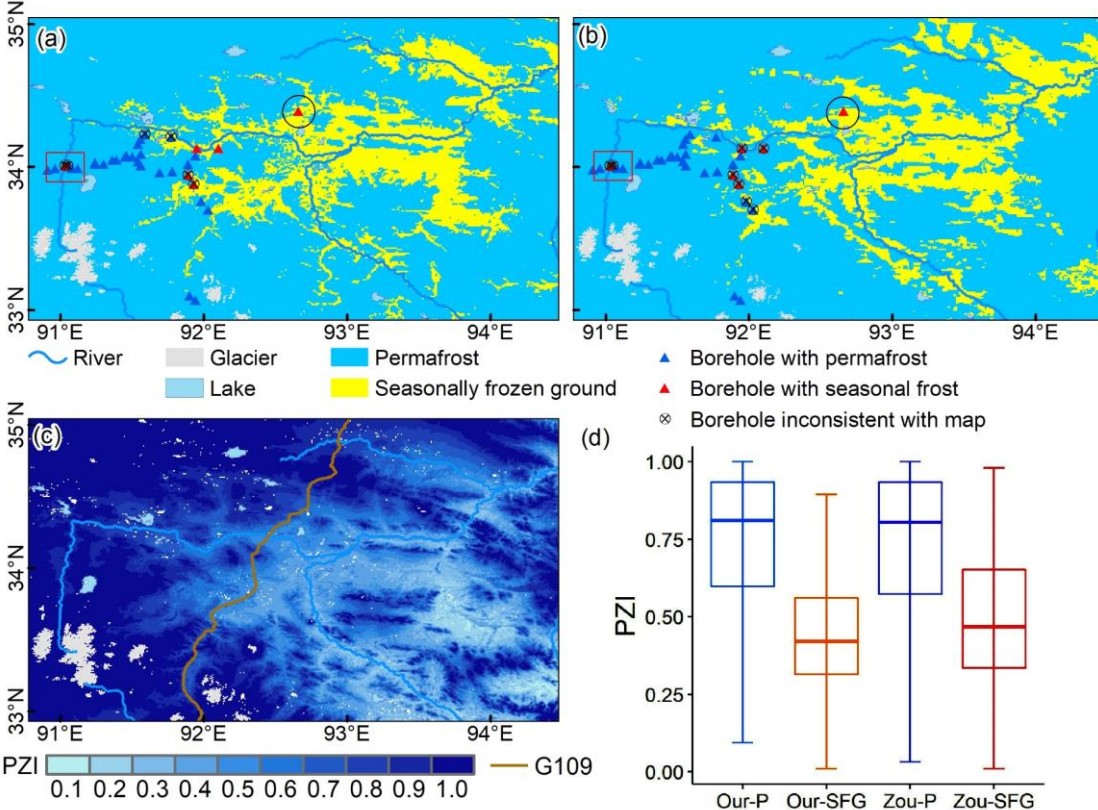

**Figure 13. Evaluation of (a) our map and (b) Zou map in the headwaters of the Yangtze River (region c in Fig. 11) based on**
**borehole observations and permafrost zonation index (PZI) in this region. (c) shows the spatial distribution of PZI in the region,**
**and (d) the statistical distributions of PZI in permafrost and SFG zones. The borehole shown in black circle was drilled around**
**2010 (Zhao et al., 2021), while the others were drilled in 2020 during the Second Tibetan Plateau Scientific Expedition (Li et al.,**
**2022). For (d), the same boxplot notations apply as in Fig. 8.**

Similar to the Yangtze River source areas, the Yellow River source areas (region d in Fig. 11, Fig. 14) also have
considerable discrepancies between our map and the Zou map. In this region, 7 boreholes from a previous study (Li et al.,
2016) were used as an independent reference. Based on these borehole observations, our map is shown to be more accurate,
as 5 of the 7 borehole locations are identified correctly in our map, but only 3 in the Zou map. In addition, considering that
elevation is the main factor controlling the permafrost distribution in this region, we referred to an analysis of elevation-
related characteristics in this region. According to a previous field study (Li et al., 2016), the lower limit of permafrost
occurrence in this region is around 4300 m. Our map displays more consistency of permafrost distribution in line with
elevational characteristics than the Zou map, and the boundaries of permafrost zones of our map extend along the 4300 m
contour in this region (Fig. 14). The Zou map obviously exaggerates permafrost areas near the two lakes where elevations
are lower than 4300 m a.s.l.

The improved performance of our map may benefit from the increased accuracy of model inputs, consideration of local
factors, and full exploitation of the survey-based subregion permafrost map. Therefore, as a better estimation of permafrost



distribution on the QTP in 2010, our map could serve as an appropriate benchmark map as well as a feasible constraint for land surface modeling on the QTP.

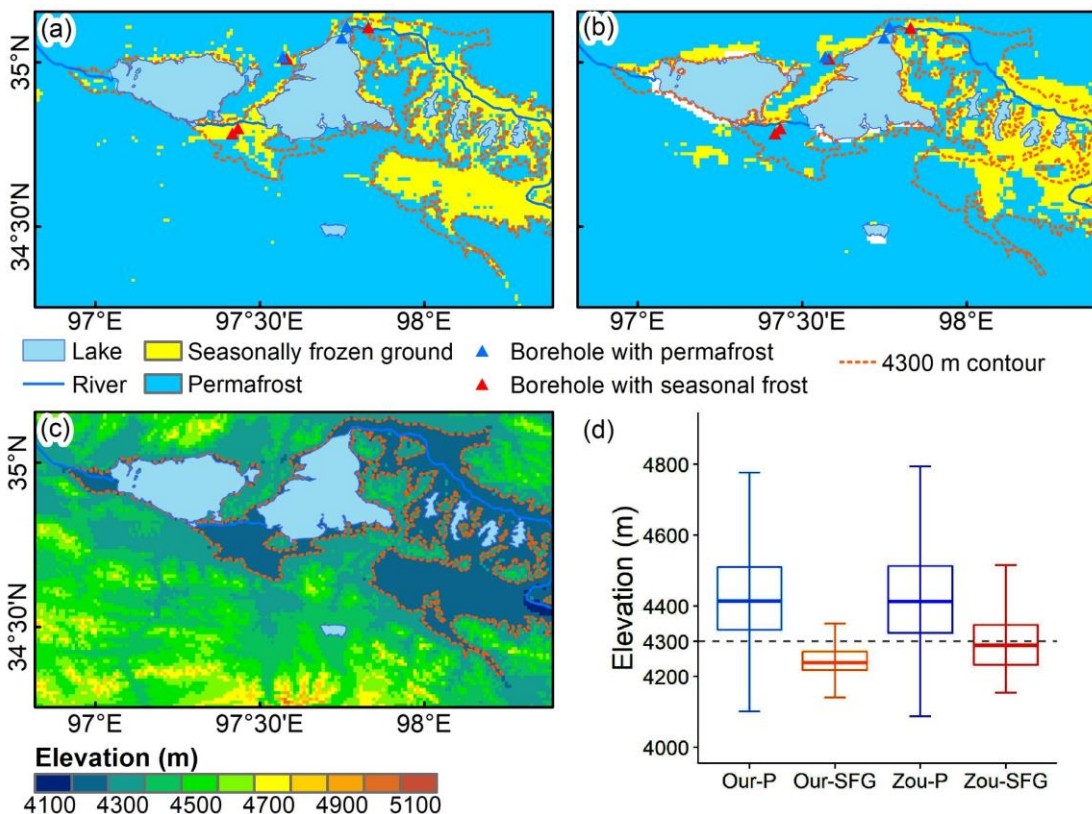


**Figure 14. Enlarged maps of the permafrost distribution in the Yellow River source areas (region d in Fig. 11) in (a) our map and (b) the Zou map along with (c) the elevation distribution. (d) shows the statistical distribution of elevations in the permafrost and SFG zones in both maps. The same boxplot notations apply to (d) as in Fig. 8.**

## 5 Data availability

The new 2010 permafrost distribution map and associated data (annual DDT and DDF data derived from MODIS LST data and soil clusters over the Qinghai-Tibet Plateau) are available at the repository hosted in figshare (Cao et al., 2022): https://doi.org/10.6084/m9.figshare.19642362. The dataset is provided as GeoTIFF files (.tif).

## 6 Conclusions

        This study provides a permafrost distribution map with a spatial resolution of 1-km over the QTP in 2010 using an
effective mapping approach, namely FROSTNUM/COP (Hu et al., 2020). This approach estimates permafrost distribution

through a semi-physical model based on satellite temperature data and accounts for the effects of local factors by fully utilizing survey-based subregion permafrost maps as constraints. Input ground surface thawing and freezing indices (relative error < 10%) were obtained from interpolated all-weather MODIS LST data. The preexisting equifinality problem of this approach has been well mitigated by including boundary consistency as part of the objective function in optimization and

uncertainties in the resulting map were further reduced by an ensemble simulation of this approach.

According to the new 2010 map, excluding glaciers and lakes, permafrost underlies about $1.086 \times 10^6$ km$^2$ (41.2% of the total QTP area), and seasonally frozen ground underlies about $1.447 \times 10^6$ km$^2$ (54.9% of the total QTP area) on the QTP in 2010. Permafrost extends continuously across the Qiangtang Plateau in the north-central QTP. Seasonally frozen ground is mainly distributed in the south and east QTP. Our map also reveals that seasonally frozen ground has appeared widely in the

river source areas in the eastern QTP due to multifaceted effects of low latitude and local factors such as thermal erosion of rivers.

This map has good consistency with the survey-based subregion permafrost maps, with a Kappa coefficient of 0.74 much higher than that of two recently published maps (Zou et al., 2017; Wang et al., 2019). When validated against 72 borehole observations of permafrost presence collected around 2010, we concluded the better performance of our map, for

which the overall accuracy is 0.85 and the Kappa value is 0.43, while the Kappa values for the Zou map and Wang map are 0.30 and 0.14, respectively. We also conducted comprehensive performance evaluations of our map and the Zou map (Zou et al., 2017), the latter which has been widely used as a reference map in existing studies, in typical regions with distinguishable differences between the two maps, based on various indicative information on the presence/absence of permafrost, such as satellite images, PZI statistics, elevation-related characteristics, as well as observations from extra

independent boreholes. Based on all these evidences, our map shows more acceptable characteristics of frozen ground distribution in those typical regions than the Zou map.

The new 2010 permafrost distribution map can thus provide accurate and fundamental information for QTP permafrost and can serve as a benchmark map of sufficient quality to calibrate/validate spatial simulations of land surface models on the QTP and as a historical reference when projecting future changes of QTP permafrost.

**Author contributions.**

Conceptualization, Z.N.; methodology, Z.N., Z.C., J.H. and Y.C.; formal analysis, Z.C.; writing—original draft, Z.C. and Z.N.; writing—review and editing, Z.N., Z.C. and Y.C.; supervision, Z.N.; funding acquisition, Z.N. and Y.Z.. All authors have read and agreed to the published version of the manuscript.

**Competing interests.**

The contact author has declared that neither they nor their co-authors have any competing interests.



**Acknowledgements.**

We would like to thank NASA (https://search.earthdata.nasa.gov), National Tibetan Plateau Data Center (https://data.tpdc.ac.cn), the National Cryosphere Desert Data Center (https://www.ncdc.ac.cn), and Science Data Bank (https://www.scidb.cn) for providing the data. We also thank the scientists who participated in the Second Tibetan Plateau 720 Scientific Expedition and Research campaign for their hard work in collecting valuable in situ data on the QTP.

**Financial support.**

This research has been financially supported by National Natural Science Foundation of China (grant nos. 41971074 and 42171125). Zetao Cao also received the support from the Postgraduate Research & Practice Innovation Program of Jiangsu Province (KYCX22_1583).

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
