# Peer review of "A new 2010 permafrost distribution map over the Qinghai-Tibet Plateau based on subregion survey maps: a benchmark for regional permafrost modeling"

_Earth System Science Data, 2022_

## Author Comment (AC1)

**Responses to Reviewer #1**

Text in red are the reviewer's comments; **those in black** are the authors' replies and explanations to the reviewer's comments; and those in blue are the revised texts appeared in the revised manuscript.

Dear authors,

this paper develops a refined model for permafrost distribution on the Qinghau-Tibet Plateau (QTP) based on a prior publication in Permafrost Periglac. (Hu et al. 2020). The extensions of the model contain the introduction of further metrics (F) aiming at guaranteeing boundary consistency and ensemble simultation. The paper is very extensive w.r.t. the model description (7 pages) and discussion of the results (15+ pages). The output data, i.e. a permafrost map of QTP as well as thawing/freezing indices and soil clusters in the form of 1km grid raster files are provided as data artefacts. As such this paper is very much a methods paper and not a typical data description paper targeted for publication in ESSD. It is commendable however that this paper makes use of recently published consolidated ground surface data from QTP (Zhao 2021) for model validation.

**Response:**

Many thanks for your comments on our work. This paper aims to provide a QTP permafrost map as a potential benchmark for modeling results, with elaborate methodological descriptions and relevant discussion. A benchmark map requires very high accuracy ensured by robust, reproducible methodology and a solid data base. During the writing process, we also learned from similar papers published in ESSD (e.g. Friedl et al., 2021; Chen et al., 2021; and Ran et al., 2022).

In this revision, as requested by the reviewers, we have shortened the paper by moving descriptions of some sub-procedures that are not core to the methodology to the supplementary materials in the hope of focusing more on the map and its specific methodology.

The moved contents include 1) the introduction to the solar-cloud-satellite geometry (SCSG) based interpolation approach (lines 226-239 in the original manuscript), 2) two approaches for estimating the annual thawing index (lines 252-277 in the original manuscript) and the corresponding results (lines 386-393 in the original manuscript), and 3) the instance of 'boundary cell' to help understand our proposed concept of boundary consistency (lines 305-345 in the original manuscript).

Refs:

Chen, Y., Liang, S., Ma, H., Li, B., He, T. and Wang, Q.: An all-sky 1km daily land surface air temperature product over mainland China for 2003-2019 from MODIS and ancillary data, Earth Syst. Sci. Data, 13, 4241-4261, 10.5194/essd-13-4241-2021, 2021.

Friedl, P., Seehaus, T. and Braun, M.: Global time series and temporal mosaics of glacier surface velocities derived from Sentinel-1 data, Earth Syst. Sci. Data, 13, 4653-4675, 10.5194/essd-13-4653-2021, 2021.

Ran, Y., Li, X., Cheng, G., Che, J., Aalto, J., Karjalainen, O., Hjort, J., Luoto, M., Jin, H., Obu, J., Hori, M., Yu, Q. and Chang, X.: New high-resolution estimates of the permafrost thermal state and hydrothermal conditions over the Northern Hemisphere, Earth Syst. Sci. Data, 14, 865-884, 10.5194/essd-14-865-2022, 2022.

Concerning the refined model I have the following comments:

You extend your the model by ensemble simulations. Yet the chosen parameter of 1000 runs seems arbitrarily picked. It is unclear how the number of runs affects your solution (accuracy/performance etc). In order to judge the utility of your proposed method this should be investigated.

**Response:**

As values of parameter E were estimated by the particle swarm optimization approach and one single time of optimization could suffer from inappropriate initial values and local optima, we chose to run the parameter optimization many times and let the result be determined by the majority voting of these estimates.

However, if the number of runs is not big enough, it is still possible that most estimates of these runs are inappropriate, compromising the efficiency of majority voting. Therefore, the number of runs should be large enough. We have investigated how the mean values and standard deviations of E estimates change with the number of runs (Fig. R1). The mean values of E become stable rapidly when the number of runs is over 100. While the standard deviations of eight soil clusters all become stable when the number of runs is over 700. It indicates that 700 runs could be enough to produce a suitable set of estimates for majority voting. Therefore, we chose 1000 times runs in this study.

We then investigated how the number of runs affects the final map. We produced permafrost maps voted by one to ten hundred runs, and find there are only very minor differences (<0.005%) between them. Though 100 seems enough, we still choose a larger number of runs (e.g. 1000) in our study to ensure stable standard deviations of E estimates (Figure R1).

The above discussion has mentioned in the manuscript briefly.

[Figure]

**Figure R1. Changes in mean values and standard deviations of soil parameter E along with number of parameter optimization runs (not shown in the revised manuscript)**

The distribution of ground control data used (weather stations/GST) in QTP is very inhomogeneous with many stations in the East and few in the west. This is nicely illustrated in (Zhao 2021). However your model approahc seems to not take this into special consideration. How do you deal with very sparse ground control data?

**Response:**

As a fact, the observation sites on the QTP are indeed unevenly distributed, and this situation is quite inevitable due to the harsh environment, which prevents the observation of more ground control data, especially in the west QTP regions. In our estimation of ground surface temperature (GST) from land surface temperature (LST), the established method, however, can effectively handle the effects of spatially inhomogeneous distribution of observation sites on the QTP.

In our approach, 0cm ground temperature (GST) measurements from weather stations were used to correct for the thermal offset between LST and GST. Based on our and others' (Wang et al., 2011) investigation, the GST-LST thermal offset is only significant on the eastern QTP during the growing season because of the vegetation cover. In contrast, in the western QTP where vegetation cover is rather low (NDVI < 0.1) (Fig. R2) or in seasons other than the growing season, the thermal offset is almost negligible. This is why we didn't carry out a correction for the freezing index. Our multilinear regression model established to correct for the thermal offset is a function of NDVI and latitude, predicting a small thermal offset when NDVI is small. Therefore, our model can effectively work for both eastern and western QTP even though the relationship is built from sites mostly on eastern QTP.

What's more, though thermal offsets between GST and LST cannot be fully eliminated through the regression model, the effects of residual offsets can be further reduced in the optimization phase of our approach, as the effects of GST-LST thermal offsets can be compensated by adjusting values of soil parameter $E$ to achieve a best possible agreement with the survey-based subregion permafrost maps. Two survey-based subregion permafrost maps (West Kunlun and Gaize) can represent the environmental conditions of permafrost in the western QTP.

We improved relative descriptions on how this method works also for western QTP where few weather sites are available.

[Figure]

**Figure R2. Map showing the normalized difference vegetation index (NDVI) distribution on the Qinghai-Tibet Plateau, the locations of meteorological stations. (not shown in the revised manuscript)**

Ref:

Wang, Z., Nan, Z., and Zhao, L.: The applicability of MODIS land surface temperature products to simulating the permafrost distribution over the Tibetan Plateau. Journal of Glaciology and Geocryology, 33(1), 132-143 ,2011. (in Chinese)

There exist a number of other models to estimate permafrost (extent) and thermal regime covering e.g. the whole northern hemisphere and available at 1km grid cell size: Youhua Ran, Xin Li, Guodong Cheng, Jingxin Che, Juha Aalto, Olli Karjalainen, Jan Hjort, Miska Luoto, Huijun Jin, Jaroslav Obu, Masahiro Hori, Qihao Yu, and Xiaoli Chang: New high-resolution estimates of the permafrost thermal state and hydrothermal conditions over the Northern Hemisphere. https://doi.org/10.5194/essd-14-865-2022. How does your approach and results derived compare to these "global" models? Especially since when considering the whole northern hemisphere much more ground data exists?

**Response:**

Our study aims to provide a historical reference for permafrost simulation studies. With this a goal, we believe that our map and the map of Ran et al. (2022) (hereinafter, Ran map) are not entirely comparable for the following reasons.

First of all, to provide a historical reference of 2010, the forcing data in our study are all from 2005 to 2010, but 99% of measurements used to produce Ran map were made during 2000-2016 (Ran et al., 2022). It means that Ran map may represent an averaged permafrost status of a longer period of time, which is not intended to serve as a historical reference. In addition, Ran map is estimated by the ensemble mean of 1000 runs of four machine learning methods trained by in-situ MAGT and ALT, which is

methodologically similar to the Wang map (Wang et al., 2019) (We compared ours with Wang map in our study). Such mapping methods can be satisfactory in circum-Arctic areas (e.g. Siberian and north Canada), because in these areas, evenly-distributed in-situ observations are available and land surface is more homogeneous than Plateaus. But on the QTP, machine learning based approaches may not be that effective due to inadequate ground data and insufficient consideration of local factors.

We have downloaded the dataset of Ran et al. (2022) and compared the permafrost extent (defined as permafrost probability > 0 according to fig.7 in Ran et al. (2022)) with our map and other existing QTP permafrost maps cited in our study (Fig. R3, Table R1, Table R2). Compared with survey-based permafrost maps and borehole observations, Ran map has similar a performance to Wang map which is also based on machine learning methods. Both of them overestimated permafrost extent on the QTP.

Considering there are much more ground data exists in the whole northern hemisphere (especially in circum-Arctic areas), we indeed plan to extend our approach to mapping the permafrost distribution in the whole northern hemisphere.

We have no plan to include Ran map in our manuscript as a comparison reference for our map, as ours and Ran map were created for different purposes.

[Figure]

**Figure R3. Spatial distributions of frozen ground in subregions from (a) survey-based maps, (b) our map, (c) the map from Zou et al. (2017), (d) the map from Wang et al. (2019), and (e) the map from Ran et al. (2022). Triangle symbols mark the locations of boreholes drilled around 2010 where the type of frozen ground was identified.**

**Table R1. Kappa values measured between the evaluated maps (our map, Zou map, Wang map, and Ran map) and survey-based permafrost distribution maps in the subregions.**

|  | West Kunlun | Gaize | Aerjin | G308 | Wenquan | All subregions |
|---|---|---|---|---|---|---|
| Our map | 0.62 | 0.71 | 0.71 | 0.68 | 0.70 | 0.74 |
| Zou map | 0.63 | 0.48 | 0.38 | 0.46 | 0.65 | 0.55 |
| Wang map | 0.63 | 0.38 | 0.00 | 0.68 | 0.46 | 0.50 |
| Ran map | 0.36 | 0.32 | 0.01 | 0.49 | 0.49 | 0.39 |

**Table R2. Measures of confusion matrices describing the performance of the evaluated maps (our map, Zou map, Wang map, and Ran map) at the borehole locations. To fit the binary classification, permafrost is regarded as positive and seasonally frozen ground negative. n=72.**

|  | Our map | Zou map | Wang map | Ran map |
|---|---|---|---|---|
| True positives (rate) | 55 (90.2%) | 53 (86.9%) | 61 (100.0%) | 61 (100%) |
| False positives (rate) | 5 (45.5%) | 6 (54.5%) | 10 (90.9%) | 8 (72.7%) |
| True negatives (rate) | 6 (54.5%) | 5 (45.5%) | 1 (9.1%) | 3 (27.3%) |
| False negatives (rate) | 6 (9.8%) | 8 (13.1%) | 0 (0.0%) | 0 (0%) |
| Accuracy | 84.7% | 80.6% | 86.1% | 84.7% |
| Cohen's Kappa | 0.43 | 0.30 | 0.14 | 0.39 |

Refs:

Ran, Y., Li, X., Cheng, G., Che, J., Aalto, J., Karjalainen, O., Hjort, J., Luoto, M., Jin, H., Obu, J., Hori, M., Yu, Q. and Chang, X.: New high-resolution estimates of the permafrost thermal state and hydrothermal conditions over the Northern Hemisphere, Earth Syst. Sci. Data, 14, 865-884, 10.5194/essd-14-865-2022, 2022.

Wang, T., Yang, D., Fang, B., Yang, W., Qin, Y. and Wang, Y.: Data-driven mapping of the spatial distribution and potential changes of frozen ground over the Tibetan Plateau, Sci. Total Environ., 649, 515-525, https://doi.org/10.1016/j.scitotenv.2018.08.369, 2019.

Zou, D., Zhao, L., Sheng, Y., Chen, J., Hu, G., Wu, T., Wu, J., Xie, C., Wu, X. and Pang, Q.: A new map of permafrost distribution on the Tibetan Plateau, The Cryosphere, 11, 2527-2542, https://doi.org/10.5194/tc-11-

2527-2017, 2017.

You specifically mention that your dataset/model is representative for the year 2010 and tries to not incorporate all historic data but only such observations that are relevant to the specific mapping year. yet in multiple places you incorporate data or discuss processes over longer periods of time, e.g. 2005-2010. How did you arrive at this again arbitrarily picked boundary of 2005? and why not incorporate data beyond 2010?

**Response:**

In our approach, the optimization targets subregions maps based on field works conducted in 2009-2010. Freeze/thaw indices over 2005-2010 are used as forcing data to drive the model. There are several reasons why we derived freeze/thaw indices based on a short period before the target year (2010) instead of the single target year (2010).

First, permafrost is defined as the frozen ground for at least two consecutive years and buried underground, permafrost situation in a specific year can be affected by climate conditions of several years before but cannot be affected by years after, thus we didn't incorporate data beyond 2010.

Second, a previous study on the mapping approach (Hu et al., 2020) show that forcing data from five to six years of are suitable for permafrost mapping.

Third, we actually used an average climatic condition over 2005-2010, to avoid the possible influence of abnormal single-year meteorological conditions.

Last, many automatic weather stations which provide GST measurements were put into operations since 2005, so we chose a period 2005-2010.

We provided justifications in the revised manuscript regarding the selection of 2005-2010 for deriving freeze/thaw indices as forcing to our mapping approach.

Ref:

Hu, J., Zhao, S., Nan, Z., Wu, X., Sun, X. and Cheng, G.: An effective approach for mapping permafrost in a large area using subregion maps and satellite data, Permafrost Periglac., 31, 548-560, https://doi.org/10.1002/ppp.2068, 2020.

Concerning the dataset I have the following comments

You present a number of comparisons of your resulting map with other data (Zhou map). In order to continuously be able to improve this modeling work (and reuse your data) it would be helpful if the comparison data was made available as well.

**Response:**

Currently, our open dataset coming with this paper does not include these comparison data in order

to avoid possible copyright infringement. To facilitate access to these data, we created a special section in the supplementary materials to provide information on where these comparison data can be downloaded, as well as links to our open dataset.

It would be very helpful for the community if your code would be made available with this model study.

**Response:**

Yes, codes will be made publicly available on github (https://github.com/nanzt ), as well as on our research group website at https://permalab.science/publications/codes, where codes of a modified Noah LSM model associated with a previous paper (Earths Futures) were already freely available.

Similarliy it would be very helpful if you could list the exact sources of all input data used in one place, similar to your data products.

**Response:**

Thanks for the suggestion. We created a special section in supplementary materials to summarize accessible sources of all input data, comparison data, as well as resulting datasets and codes of mapping.

I have downloaded your dataset and was able to load this into QGIS.

**Response:**

Thank you for your attention to our work and dataset.

---

## Author Comment (AC2)

**Responses to Reviewer #2**

Text in red are the reviewer's comments; **those in black** are the authors' replies and explanations to the reviewer's comments; and those in blue are the revised texts appeared in the revised manuscript.

Dear authors,

This paper presented a new 2010 map of permafrost distribution for the Qinghai-Tibet Plateau (QTP) produced through a modified version of Hu et al. (2020)'s model. The model used a combination of field data from around 2010 and satellite-derived ground surface thawing/freezing indices as input, and survey-based subregion permafrost maps as constraints. They present the resulting map a the best available one as of now, which could be used in the future as a benchmark for calibration and validation of simulations. Although their goal of the paper and the methodology used to reach it are not entirely new or innovative, their approach is robust and results of high quality.

Please note that since I am not a modeller, I was not able to comment on the pertinence of the model. I could only comment on the overall quality of the paper and the scientific approach to the problem.

In general, the paper would benefit from being more concise. The description of the model is quite exhaustive and some sections could go in supplementary materials (see specific comments). I also found that the manuscript would benefit from having a results section and then a discussion rather than a Results and Discussion section together. This would facilitate reading and encourage the authors to describe their ideas/conclusions more clearly. I also suggest adding a section for "Simulation limitations", which would acknowledge the limitations of the model/study and what could not be done with the present paper.

Below you will find specific and technical comments that I hope you will find useful.

Best regards,

Samuel Gagnon

**Response:**

We are extremely grateful for your thorough comments on our work, which greatly improve the manuscript quality. In the reversion, we shortened the paper by moving some sections to supplementary materials, including 1) the introduction to the solar-cloud-satellite geometry (SCSG) based interpolation approach (lines 226-239 in the original manuscript), 2) two approaches for estimating the annual thawing index (lines 252-277 in the original manuscript) and the corresponding results (lines 386-393 in the original manuscript), and 3) the instance of 'boundary cell' in parameter optimization approach (lines 305-345 in the original manuscript).

Referring to existing data papers published in ESSD, we decided not to separate the Results and Discussion sections. Most papers on ESSD of this type, like ours, have a mixed results and discussion

section. I suspect that this way readers are more likely to focus on the results and their interpretations, and a separate discussion section makes it more like a normal research paper. However, in response to your comment, we have added a simulation limitations section as subsection 4.6.

Below are the point-by-point responses.

**Specific comments:**

Line 62: Be careful with the tone when comparing with other works as sometime the adjectives you are using sound condescending. I've seen this multiple times throughout the text. Although I don't think it is intentional, sometime your choices of word is harsher than it needs to be. Previous work may seem flawed now with new technology/knowledge, but it was the best that could be done at the time, and we have to keep that in mind when evaluating past work.

**Response:**

Thank you very much for this advice. We definitely do not intend to use an inappropriate tone just because we are not native speakers and cannot properly convey our meaning. In this revision, we paid special attention to the problem you mentioned. We hope after this revision the problem you mentioned can be eliminated. Here in line 62, we decided to remove the statement "the quality of these maps is often unsatisfactory".

Line 79: Could you be more specific and refer to the works you are talking about? I don't think it's true for all works done in all areas with permafrost, so it would be good to have specific references pertinent to the QTP.

**Response:**

As far as we know about the QTP permafrost modeling studies, this is true. Until recent years spatial modeling of permafrost for the entire QTP has emerged. Some studies (e.g. Guo et al., 2012) applied land surface models to the QTP without any calibration.

We have added two references here (Qin et al., 2017; Wu et al., 2018). These two studies are about permafrost modelling using land surface models, where the models were calibrated using data only from a few sites (based on one site in the work of Wu et al., 2018 and four sites in the work of Qin et al., 2017).

Refs:

Guo, D. and Wang, H.: Simulation of permafrost and seasonally frozen ground conditions on the Tibetan Plateau, 1981-2010, Journal of Geophysical Research: Atmospheres, 118, 5216-5230, https://doi.org/10.1002/jgrd.50457, 2013.

Qin, Y., Wu, T., Zhao, L., Wu, X., Li, R., Xie, C., Pang, Q., Hu, G., Qiao, Y. and Zhao, G.: Numerical modeling of the active layer thickness and permafrost thermal state across Qinghai‐Tibetan Plateau, Journal of Geophysical Research: Atmospheres, 122, 11,604-11,620, https://doi.org/10.1002/2017JD026858, 2017.

Wu, X., Nan, Z., Zhao, S., Zhao, L. and Cheng, G.: Spatial modeling of permafrost distribution and properties on the Qinghai-Tibet Plateau, Permafrost and periglacial processes, 29, 86-99, https://doi.org/10.1002/ppp.1971, 2018.

Line 88: Do you mean that ideally a map should be based on field observations, or that that they should be specifically independent of land surface model simulations but that other types of simulations are ok? Be specific, you could remove "land sur face model" and just put simulations. Also, there's not mention of the number of observations, which is important because a map based on only a handful number of observations is unreliable. The first criterion of a good benchmark map could be changed to "a map based on a large number of robust observations independent of simulations".

**Response:**

We mean an ideal map for benchmarking model simulations must have a solid base of observations from all kinds of reliable measurements, e.g. field observations, remotely sensed observations. Thank you for your incisive comment, those observations should be independent not only of land surface models but also of any other relevant models, we have revised it accordingly.

"a map based on an adequate number of robust observations independent of simulations".

The statement in line 88 "a map based on credible observations and independent of land surface model simulations" didn't mean that an ideal map should be based on field observations. The 'observations' in the sentence include not only field observations but also other kinds of observations (e.g. remote sensing data). However, the original sentence could raise misunderstanding, and the sentence "a map based on a large number of robust observations independent of simulations" is much better.

**Response:** We have revised it as suggested.

Line 98: Before we go into the data used for mapping, I'm missing a "Study site" section where you describe the QTP. It could be brief and even integrated with another section, but I think it would be useful to have a description of the area with some baseline information about the climate (e.g., snow, summer & winter temperatures, insolation) and permafrost. In other words, mention what is typical of the QTP so that we can have a better idea of what you are simulating.

**Response:**

We have added a subsection about a brief introduction to the QTP in section 2:

The QTP (bounded within 73.5°E-104.5°E and 26°N - 40°E) has an area of about $2.6\times10^6$ km$^2$ and

is surrounded by high mountain ranges with relatively flat terrain in the interior (Fig.1). Most of the QTP lies at an elevation of 3000-5000 m a.s.l. with an average of about 4000 m a.s.l. Temperature and precipitation vary substantially over the QTP (Gerlitz et al., 2014). The mean annual air temperature ranges from -5 to 5°C in most areas with elevations between 3000-5000 m a.s.l., with the highest monthly temperature at about 10°C in July and the lowest at -10°C or below in January. In the last five decades preceding 2010, air temperature increased by about 0.3-0.4°C per decade, which is more than twice the global warming rate (Zhang et al., 2019). Mean annual precipitation decreases from more than 700mm in the southeast towards about 50mm in the northwest, and about 90% of precipitation falls during the growing season of May to September (Peng et al., 2019). The QTP is mainly covered by alpine desert, alpine meadow and forest, transitioning from the northwest to the southeast (Wang et al., 2016). Snow cover on the QTP is thin and of short duration (Wu and Zhang, 2008). The QTP has the largest permafrost distribution in the mid-latitudes, occupying about 40% of total area (Cao et al., 2019a). Ice-rich layers are often found near the permafrost table on the QTP, where the active layer thickness is generally 2-3 m (Zhao et al., 2020), much thicker than the circum-Arctic permafrost. The permafrost thickness on the QTP ranges from several meters to about 350 m while the depth of zero annual amplitude is generally between 3.5 and 17 m (Zhao et al., 2020). The QTP permafrost is also characterized by a high mean annual ground temperature, which is higher than -3°C in most permafrost regions (Zhao et al., 2020).

Refs:

Cao, B., Zhang, T., Wu, Q., Sheng, Y., Zhao, L. and Zou, D.: Brief communication: Evaluation and inter-comparisons of Qinghai-Tibet Plateau permafrost maps based on a new inventory of field evidence, The cryosphere, 13, 511-519, https://doi.org/10.5194/tc-13-511-2019, 2019a.

Gerlitz, L., Conrad, O., Thomas, A. and Böhner, J.: Warming patterns over the Tibetan Plateau and adjacent lowlands derived from elevation-and bias‐corrected ERA-Interim data, Clim. Res., 58, 235-246, https://doi.org/10.3354/cr01193, 2014.

Peng, S., Ding, Y., Liu, W. and Li, Z.: 1 km monthly temperature and precipitation dataset for China from 1901 to 2017, Earth Syst. Sci. Data, 11, 1931-1946, https://doi.org/10.5194/essd-11-1931-2019, 2019.

Wang, Z., Wang, Q., Zhao, L., Wu, X., Yue, G., Zou, D., Nan, Z., Liu, G., Pang, Q. and Fang, H.: Mapping the vegetation distribution of the permafrost zone on the Qinghai-Tibet Plateau, J Mt. Sci., 13, 1035-1046, https://doi.org/10.1007/s11629-015-3485-y, 2016.

Wu, Q. and Zhang, T.: Recent permafrost warming on the Qinghai‐Tibetan Plateau, Journal of Geophysical Research: Atmospheres, 113, 1-22, https://doi.org/10.1029/2007JD009539, 2008.

Zhang, G., Nan, Z., Wu, X., Ji, H. and Zhao, S.: The role of winter warming in permafrost change over the Qinghai‐Tibet Plateau, Geophys. Res. Lett., 46, 11261-11269, https://doi.org/10.1029/2019GL084292, 2019.

Zhao, L., Zou, D., Hu, G., Du, E., Pang, Q., Xiao, Y., Li, R., Sheng, Y., Wu, X. and Sun, Z.: Changing climate and

the permafrost environment on the Qinghai-Tibet (Xizang) plateau, Permafrost and Periglacial Processes, 31, 396-405, https://doi.org/10.1002/ppp.2056, 2020.

Line 99: I think section 2.1 could be longer and have more information about the study by Zhao et al., 2017 since it's in Chinese. It would be good to have more information about the maps, i.e., their resolution and how exactly they were created (e.g., just field observations? Mapping software/algorithm?). It doesn't have to be too long, but in its current form I feel that we are missing so key information about those reference maps.

**Response:**

Information on the mapping methods and source data for these survey-based subregion maps was added to the revised manuscript. In the Wenquan and West Kunlun subregions, permafrost distribution was mapped using the multivariate adaptive regression splines (MARS) model trained on large samples of field measurements: 130 ground penetrating radar profiles and 21 boreholes in Wenquan, and 103 ground penetrating radar profiles, 50 pits and 13 boreholes in WestKunlun.

In the Gaize, Aerjin and G308 subregions, permafrost distribution was firstly investigated using 14 ground penetrating radar profiles, 20 ground data processing profiles and 22 boreholes, and then mapped based on relationships between altitude limits of permafrost occurrence and topographic features (Chen et al., 2016).

Ref:

Chen, J., Zhao, L., Sheng, Y., Li, J., Wu, X., Du, E., Liu, G. and Pang, Q.: Some characteristics of permafrost and its distribution in the Gaize area on the Qinghai—Tibet Plateau, China, Arctic, Antarctic, and Alpine Research, 48, 395-409, http://dx.doi.org/10.1657/AAAR0014-023, 2016.

Line 101: what is the "surroundings" of permafrost? Consider removing it

**Response:** Removed.

Line 101: Representative of what? Of the QTP? Please specify

**Response:**

This sentence has been revised to: Intensive surveys were conducted in five areas characterized by distinct climatic and geographic conditions, as representatives of the diverse permafrost environments on the QTP.

Line 102: The tone of the sentence is too engaged in my opinion. I've seen it a couple of times in the paper, be careful to remain formal/informative, which is more typical of scientific writing.

I suggest changing the sentence to "Comprehensive information was acquired through field observations, mechanical excavation, geophysical exploration techniques (ground penetrating radar, time-domain electromagnetic), and borehole drilling, which allowed to map the permafrost distributions with high precision in all five subregions."

**Response:**

Many thanks for correcting the sentence and it was accepted.

Line 110: I suggest that you revise the color schemes of the permafrost maps. Shades of purple have been used in other maps (e.g., Obu et al., 2019, https://doi.org/10.1016/j.earscirev.2019.04.023), which is very useful if you want to include bodies of water and avoid blue shades. Also, with the scale of the maps (other ones in the paper), it is difficult to differentiate glacier and water bodies, consider changing the colors. It would be nice if the colors of the topographic map and the permafrost maps did not repeat.

**Response:**

We have changed the colors of lakes and glaciers on all maps in the paper to make them more conspicuous, and we also adjusted the elevation color map to avoid repeating colors in the topographic map and permafrost maps. However, we prefer not to use shades of purple to represent permafrost, because a key characteristic of permafrost is that it contains ground ice, and a cool color like blue seems more suitable for representing permafrost.

[Figure]

**original figure 1**

[Figure]

**Figure 1 (revised). Map showing the topography of the Qinghai-Tibet Plateau (QTP), the locations of meteorological stations, and the subregions with extensive field investigation. Inset maps show the local permafrost distributions in the five subregions based on the survey data circa 2010. WK: West Kunlun; GZ: Gaize; AJ: Aerjin; WQ: Wenquan.**

Line 121: I find the acronyms DDT and DDF confusing. DDT should be TDD for Thawing Degree-Days, and DDF should be FDD for Freezing Degree-Days. In addition, TDD and FDD don't denote the freezing and thawing indices. While degree-days are the departure of the mean daily temperature from a reference temperature (0°C), the thawing index (Ti) and freezing index (Fi) are the sum of degree-days for the thawing and freezing seasons, respectively. This should be explained in the text, i.e., what are TDD and FDD, and how they are used to calculate the freezing and thawing indices (use different acronyms, e.g., Ti and Fi). Change the acronyms in the rest of the text accordingly. You kind of mention what are FDD and TDD at lines 245 and 247, so remove it once you make the change here.

**Response:**

We thank you for this suggestion, but we would like to keep the acronyms (DDT/DDF) to respect the original frost number model proposed by Nelson and Outcalt (1987), where DDT/DDF stood for thawing/ freezing index, respectively. We realized we have confused thawing/freezing index with degree-days in the original manuscript, and we made corrections accordingly.

To avoid potential misunderstanding, we revised the sentence to clarify the definition of DDT/DDF

here:

These observations were used to estimate annual ground surface thawing (DDT) and freezing indices (DDF) driving the mapping approach. DDT and DDF are defined as the cumulative number of degree-days when ground surface temperatures (GST) are above and below zero degrees Celsius, respectively (Nelson and Outcalt, 1987).

Ref:

Nelson, F.E. and Outcalt, S.I.: A Computational Method for Prediction and Regionalization of Permafrost, Arctic and Alpine Research, 19, 279-288, https://doi.org/10.1080/00040851.1987.12002602, 1987.

Line 121: "which in theory are aggregated from the 0 cm ground temperature", actually, Fi and Ti can be calculated for any depth, so you have to specify that for your study, you used the 0 cm, so remove "in theory". This sentence needs to be adjusted with the redefinition of the thawing and freezing indices.

**Response:**

We removed "in theory" and revised the sentence to clarify the calculation of thawing and freezing indices in our study. Now we defined DDT/DDF as ground surface thawing and freezing indices, specific to 0cm depth.

Line 127 to 130: You use the NDVI for estimating the vegetation conditions over the study area and I agree with that approach. However, you assume that the NDVI is a perfect match for vegetation cover over your study area, which it inherently isn't. For instance, when was the product obtained? Does that impact the result? Was cloud cover an issue? Are there specific areas that are not represented well? I suggest that you add the limitations or at least that you acknowledge that there could be some in the use of the NDVI. Conversely, you could also explain why you think this is a good proxy on the QTP (is tundra represented well?).

**Response:**

We used NDVI in two ways in this study. NDVI is used as one of the clustering variables for spatial clustering of soils, and NDVI is used as a predictor in a multilinear model to estimate GST from LST. We know NDVI may not perfectly represent the vegetation cover on the QTP. Some existing studies like Wang et al. (2016) used NDVI to characterize grassland dynamics on the QTP. While NDVI on the QTP won't face the saturation issue (because the majority of QTP is covered by alpine steppe, alpine meadow, and alpine desert) (Huete et al., 1997), NDVI in very low vegetation areas is sensitive to soil background. But for desert areas, we prefer to group them into one cluster, and regard them as almost bare lands with minor thermal offset in estimating GST.

The NDVI data we used are the composite 16-day 1-km normalized difference vegetation index (NDVI) product (MOD13A2), with no data gap. In the case of clustering, 2005-2010 averaged daily NDVIs are used as a proxy to represent spatial heterogeneity of vegetation conditions on the QTP.

We amended the text based on above explanations.

Refs:

Huete, Alfred0 R., HuiQing Liu, and Wim JD van Leeuwen. "The use of vegetation indices in forested regions: issues of linearity and saturation." IGARSS'97. 1997 IEEE International Geoscience and Remote Sensing Symposium Proceedings. Remote Sensing-A Scientific Vision for Sustainable Development. Vol. 4. IEEE, 1997.

Wang X, Yi S, Wu Q, Yang K, Ding Y. The role of permafrost and soil water in distribution of alpine grassland and its NDVI dynamics on the Qinghai-Tibetan Plateau. Global and Planetary Change. 2016, 147: 40-53. doi:10.1016/j.gloplacha.2016.10.014.

Line 137: Could you explain how the soil texture type data was used in the mapping approach?

**Response:**

Like NDVI, soil texture type as a local factor affect permafrost distribution and is thus used as a variable for spatial clustering in our approach. We clarified the purpose of soil texture type data in this revision.

Line 142: Having 131 meteorological stations with long-term record is fantastic. However, there is an obvious bias in the coverage in the station, as you mention in the text. Could this have affected the results? If so, how? It would be good to acknowledged it and mention it somewhere in the text.

**Response:**

As a fact, the observation sites on the QTP are indeed unevenly distributed. We used data at observation sites to calibrate GST. However, we believe the coverage bias would not seriously affect the result. In our estimation of ground surface temperature (GST) from land surface temperature (LST), the established method can effectively handle the effects of spatially inhomogeneous distribution of observation sites on the QTP.

In our approach, 0cm ground temperature (GST) measurements from weather stations were used to correct for the thermal offset between LST and GST. Based on our and one other's (Wang et al., 2011) investigation, the GST-LST thermal offset is only significant on the eastern QTP during the growing season because of the vegetation cover. In contrast, in the western QTP where vegetation cover is rather low (NDVI < 0.1) (Fig. R2) or in seasons other than the growing season, the thermal offset is almost negligible. Our multilinear regression model established to correct for the thermal offset is a function of

NDVI and latitude, predicting a small thermal offset when NDVI is small. Therefore, our model can effectively work for both eastern and western QTP even though the relationship is built from sites mostly on eastern QTP.

What's more, though thermal offsets between GST and LST cannot be fully eliminated through the regression model, the effects of residual offsets can be further reduced in the optimization phase of our approach, as the effects of GST-LST thermal offsets can be compensated by adjusting values of soil parameter $E$ to achieve a best possible agreement with the survey-based subregion permafrost maps. Two survey-based subregion permafrost maps (West Kunlun and Gaize) can represent the environmental conditions of permafrost in the western QTP.

The same concern was raised by another reviewer. We improved relative descriptions of how this method works also for western QTP where few weather sites are available.

[Figure]

**Figure R2. (repeated) Map showing the normalized difference vegetation index (NDVI) distribution on the Qinghai-Tibet Plateau, the locations of meteorological stations. (not shown in the revised manuscript)**
Ref:

Wang, Z., Nan, Z., and Zhao, L.: The applicability of MODIS land surface temperature products to simulating the permafrost distribution over the Tibetan Plateau. Journal of Glaciology and Geocryology, 33(1), 132-143 ,2011. (in Chinese)

Line 143: Please indicate what are the "standard meteorological variables".

**Response:**

Standard meteorological variables refer to measurements typically made at national-level weather stations, including air pressure, air temperature, precipitation, evaporation, relative humidity, wind speed, wind direction, sunshine hours, and 0 cm ground surface temperature in China national surface weather

stations. We have added this information to the text.

Line 150 to 155: I'm not familiar with permafrost conditions at the QTP (hence my comment above on site description). Hence, I don't know what permafrost thickness to expect. How did you determine the presence of permafrost with two measurements, i.e., 10 and 20 m depths? Also, at what depths were the MAGT for your three categories determined? Please specify in the text.

**Response:**

General information about QTP permafrost thickness are now provided in a "study area" section: "The permafrost thickness on the QTP ranges from several meters to about 350 m while the depth of zero annual amplitude in ground temperature is generally between 3.5 and 17 m (Zhao et al., 2020)".

For your reference here, we cited more detail information about QTP permafrost from Zhao et al. (2020): "The permafrost thickness ranges from several meters to about 350 m. Permafrost is thicker on the Qiangtang Plateau and the Kunlun Mountains which lie in the central QTP. Permafrost thickness is more than 200 m near the mountain ridges at high elevation (generally above 5,500 m a.s.l.), 60-130 m in hilly landscapes, and less than 60 m in valley bottoms on the high plateau".

MAGT is a permafrost term referring to the soil temperature at a certain depth where annual amplitude of oscillation is zero (DZAA, depth of zero annual amplitude). The dataset from Zhao et al. (2021) only provided annual-average ground temperature values at 10 m and 20 m depths, they are not the exact MAGT. MAGT is often used to indicate the thermal state of permafrost. We misused MAGT here and corrected it accordingly (10/20-m soil temperature).

The presence/absence of permafrost can be judged at the base of the active layer (the deepest depth seasonal thawing can penetrate in permafrost areas) and the depths downward. If the soils are found frozen at those depths, this location is judged as permafrost underlies. The thickness of active layer is generally 2-3m on the QTP. BTW, those borehole locations were roughly determined based on prior information before drilling was actually made.

10-20m temperature measurements are used to monitor thermal states of the soils. In the permafrost sites among those sites, DAZZ is estimated between 10-20 m, therefore, we can classify permafrost thermal stability (stable, unstable) on those sites/boreholes.

We revised the text accordingly.

Refs:

Zhao, L., Zou, D., Hu, G., Du, E., Pang, Q., Xiao, Y., Li, R., Sheng, Y., Wu, X. and Sun, Z.: Changing climate and the permafrost environment on the Qinghai-Tibet (Xizang) plateau, Permafrost and Periglacial Processes, 31, 396-405, https://doi.org/10.1002/ppp.2056, 2020.

Zhao, L., Zou, D., Hu, G., Wu, T., Du, E., Liu, G., Xiao, Y., Li, R., Pang, Q. and Qiao, Y.: A synthesis dataset of

permafrost thermal state for the Qinghai‒Tibet (Xizang) Plateau, China, Earth Syst. Sci. Data, 13, 4207-4218, https://doi.org/10.5194/essd-13-4207-2021, 2021.

Line 168 to 174: This sentence is too long, and each map needs to be explained separately. For the Zou map, was it developed using the TTOP model? The phrasing makes it unclear. You could remove "Glaciers, Frozen Ground and Deserts in China" and just put the refence to shorten it.

**Response:**

We have separated the long sentence into three short sentences: (1) "To better evaluate the new map generated in this study, two existing permafrost distribution maps with 1 km resolution representing permafrost distribution on the QTP around 2010 were used." (2) "One is a new permafrost distribution map on the Tibetan Plateau by Zou et al., (2017) (hereinafter, Zou map), which uses the temperature at the top of permafrost model (TTOP) with MODIS LST data from 2003 to 2012 as input." and (3) "The other map is a data-driven permafrost map by Wang et al. (2019) (hereinafter, Wang map) using three statistical models trained on the samples from a 2006 map (Wang, 2013), whose QTP portion was mapped using a multilinear regression model (Nan et al., 2002), and the Zou map."

Refs:

Nan, Z., Li, S. and Liu, Y.: Mean annual ground temperature distribution on the Tibetan Plateau: permafrost distribution mapping and further application (in Chinese), Journal of Glaciology and Geocryology, 24, 142-148, http://www.bcdt.ac.cn/CN/Y2002/V24/I2/142, 2002.

Wang, T.: 1:4 million map of the Glaciers, Frozen Ground and Deserts in China (2006), National Tibetan Plateau Data Center[data set], https://doi.org/10.3972/westdc.015.2013.db, 2013.

Wang, T., Yang, D., Fang, B., Yang, W., Qin, Y. and Wang, Y.: Data-driven mapping of the spatial distribution and potential changes of frozen ground over the Tibetan Plateau, Sci. Total Environ., 649, 515-525, https://doi.org/10.1016/j.scitotenv.2018.08.369, 2019.

Zou, D., Zhao, L., Sheng, Y., Chen, J., Hu, G., Wu, T., Wu, J., Xie, C., Wu, X. and Pang, Q.: A new map of permafrost distribution on the Tibetan Plateau, The Cryosphere, 11, 2527-2542, https://doi.org/10.5194/tc-11-2527-2017, 2017.

Line 189: "using techniques […]", is this for the model parameter E or for the model in general? Clarify

**Response:**

The techniques we used to estimate E include spatial clustering and parametric optimization. We clarified it as requested.

Line 190: "such as", this suggest an example, does it mean that there are more techniques than those listed in the text? If so, it would be better to avoid "such as" and list all of the techniques.

**Response:**

There are no other techniques but spatial clustering and parametric optimization, we have removed "such as".

Line 242: A reference to snow cover on the QTP in a "site description" section would be great, see comment for line 98.

**Response:**

A "study area" section has been added to the revised manuscript. We also provide a brief description of snow cover on the QTP there.

Lines 226 to 239: Consider putting this section in supplementary material.

**Response:** Done as suggested.

Line 269: You used the 131 weather stations for evaluating the performance for the interval-based estimation only or also the one-year estimation? The first and second sentence makes this confusing

**Response:**

We used the 131 weather stations for both methods. We have revised the sentence: To compare the effectiveness of the 'interval-based estimation' method and the 'one-year estimation' method, we randomly divided the 131 weather stations into a training set (70%) and a testing set (30%) for 100 times. This part about the detailed description of two estimation methods has been moved to supplementary materials.

Lines 278 to 286: Is this just an explanation of what Hu et al. (2020) have done? If so, this could be removed or put in supplementary material.

**Response:**

This is a brief introduction to the approach of Hu et al. (2020), on which we made extensions. We had a discussion about this and decided to keep this part. It is necessary so that readers can understand the basic idea of the approach (which is the same for ours and Hu's) without referring to the supplemental materials.

Lines 283 to 286: Verb tense to revise.

**Response:** Done.

Line 308: How was $\kappa$ determined? You mention it for $\beta$, but not the other, unless I missed it.

**Response:**

$\kappa$ is the Kappa coefficient calculated between the simulated map and the survey-based map. We mentioned it in "Cohen's Kappa coefficient (Cohen, 1960) between the simulation map and the survey-based subregion permafrost distribution maps was used as the only objective function." and "we retained Kappa coefficient ($\kappa$) and imposed a more stringent constraint on the objective function by adding a specially defined boundary consistency."

Ref:

Cohen, J.: A coefficient of agreement for nominal scales, Educ. Psychol. Meas., 20, 37-46, https://doi.org/10.1177/001316446002000104, 1960.

Lines 325 to 345: This section should be in supplementary material, and the paragraph at line 346 should be added to the paragraph ending at line 324.

**Response:** Done

Line 343: Can you add a reference to the C5.0 decision tree method?

**Response:**

A reference (Kuhn and Johnson, 2013) about C5.0 decision tree has been added.

Ref:

Kuhn, M. and Johnson, K.: Applied predictive modeling, Springer, 2013.

**Response:**

We specifically chose the Zou map for specific comparisons for the following reasons. Firstly, Zou map is simulated using TTOP model which is also an empirical model like the extended ground surface frost number model used in our study. Zou et al. (2017) used MODIS LST data from 2003-2012 as drivers, the same satellite data but in a short period (2005-2010) used in our mapping approach. More importantly, Zou map has been used as a reference of the present QTP permafrost distribution in many studies (Hu et al., 2019; Song et al., 2020; Mu et al., 2020; Ni et al., 2021; Yin et al., 2021). Cao et al. (2019a) regarded the Zou map as the best performing permafrost map on the QTP in an evaluation based on an inventory of field evidence. Therefore, a comparison between our map and Zou map can illustrate the effectiveness of our mapping approach.

The above information about Zou map was provided in the data section where Zou map is introduced

and we revised it by stressing the reasons why we chose it for comparison.

Refs:

Cao, B., Zhang, T., Wu, Q., Sheng, Y., Zhao, L. and Zou, D.: Brief communication: Evaluation and inter-comparisons of Qinghai-Tibet Plateau permafrost maps based on a new inventory of field evidence, The cryosphere, 13, 511-519, https://doi.org/10.5194/tc-13-511-2019, 2019.

Hu, G., Zhao, L., Li, R., Wu, X., Wu, T., Zhu, X., Pang, Q., Yue Liu, G., Du, E. and Zou, D.: Simulation of land surface heat fluxes in permafrost regions on the Qinghai-Tibetan Plateau using CMIP5 models, Atmos. Res., 220, 155-168, https://doi.org/10.1016/j.atmosres.2019.01.006, 2019.

Mu, C., Abbott, B.W., Norris, A.J., Mu, M., Fan, C., Chen, X., Jia, L., Yang, R., Zhang, T. and Wang, K.: The status and stability of permafrost carbon on the Tibetan Plateau, Earth-Sci. Rev., 211, 103433, https://doi.org/10.1016/j.earscirev.2020.103433, 2020.

Ni, J., Wu, T., Zhu, X., Hu, G., Zou, D., Wu, X., Li, R., Xie, C., Qiao, Y. and Pang, Q.: Simulation of the present and future projection of permafrost on the Qinghai-Tibet Plateau with statistical and machine learning models, Journal of Geophysical Research: Atmospheres, 126, e2020JD033402, https://doi.org/10.1002/essoar.10503593.1, 2021.

Song, C., Wang, G., Mao, T., Dai, J. and Yang, D.: Linkage between permafrost distribution and river runoff changes across the Arctic and the Tibetan Plateau, Science China Earth Sciences, 63, 292-302, https://doi.org/10.1007/s11430-018-9383-6, 2020.

Yin, G., Niu, F., Lin, Z., Luo, J. and Liu, M.: Data-driven spatiotemporal projections of shallow permafrost based on CMIP6 across the Qinghai-Tibet Plateau at 1 km2 scale, Adv. Clim. Chang. Res., 12, 814-827, https://doi.org/10.1016/j.accre.2021.08.009, 2021.

Zou, D., Zhao, L., Sheng, Y., Chen, J., Hu, G., Wu, T., Wu, J., Xie, C., Wu, X. and Pang, Q.: A new map of permafrost distribution on the Tibetan Plateau, The Cryosphere, 11, 2527-2542, https://doi.org/10.5194/tc-11-2527-2017, 2017.

Line 371: Can you give an example of how satellite images can provide indicative landscape evidence of permafrost existence?

**Response:**

Some periglacial landforms like polygons or rock glaciers (e.g. Hassan et al. 2021) visible from satellites could be indicators of permafrost. Recent studies also found evidence from aerial images to indicate permafrost degradation (e.g. Morino et al. 2019).

Refs:

Hassan J, Chen X, Muhammad S, Bazai N A. Rock glacier inventory, permafrost probability distribution

modeling and associated hazards in the Hunza River Basin, Western Karakoram, Pakistan. Science of The Total Environment. 2021, 782: 146833. doi:https://doi.org/10.1016/j.scitotenv.2021.146833.

Morino C, Conway S J, Sæmundsson Þ, Helgason J K, Hillier J, Butcher F E G, Balme M R, Jordan C, Argles T. Molards as an indicator of permafrost degradation and landslide processes. Earth and Planetary Science Letters. 2019, 516: 136-147. doi:10.1016/j.epsl.2019.03.040.

Line 379 to 381: Is it possible to cut this sentence or divide it in two? It's hard to read. In addition, the "good consistency" followed by the "despite considerable discrepancies" could be separated into two sentences, which would make it less confusing when reading, something like "however, there were considerable differences in absolute values".

**Response:** Accepted.

Lines 384 and 385: This explanation and references should reflect what is presented in the "Study site" section (see line 98 comment)

**Response:** Done. We have added a study area section echoing this part.

Line 386: You don't need to describe the figure/table in the text. I've seen a couple of time in the text (e.g., line 403). I personally think it's redundant with the figure/table caption and adds unnecessary length to the text. You can directly start talking about your observations from the figure/table (like in line 394). To correct in other places where you talk about figures/tables.

**Response:**

We have deleted redundant descriptions throughout the manuscript. BTW, this paragraph has been moved to supplementary materials.

Lines 386 to 393: In this paragraph, you describe almost the entirety of table 1, except for the ranges. It makes me wonder if table 1 is indeed needed, you could just add the ranges in the text. Something to consider.

**Response:**

We have removed the exact metrics values in the text and kept Table 1 because a table can be clearer than just words.

Lines 411 and 412: I don't know where the Qaidam Basin or the Qiangtang Plateau are, please specify or indicate in one of the maps prior to Figure 9.

**Response:** Done. We have marked them on figure 1.

Line 417: Figure 4: Please split the first sentence, e.g., "Observed discrepancies between annual thawing degree-days aggregated from in situ and interpolated data. The data was obtained from MODIS LST […]".

**Response:** We rephrase it for clarity as follows:

[Figure]

**Figure 3 (original figure 4). Thawing/freezing indices calculated from interpolated MODIS LST data (raw LST-derived DDT/DDF) and in situ observations of ground surface temperature (in-situ DDT/DDF) at each QTP weather station. The ordinate indicates the annual thawing/freezing indices averaged over the period 2005 to 2010; and the abscissa shows 131 weather stations available on the QTP.**

Line 423: Table 1: Potentially put in supplementary material (see earlier comment). If you leave the table in the manuscript, remove "The training sets consist of 70% of the available stations, and the metric values provided were calculated over the remaining 30% as testing stations.", it's redundant with the information in the text. For the ranges, please change ~ to -.

**Response:**

Done as suggested, and we put Table 1 and the corresponding text in supplementary materials now. Also, we changed the character '~' to '-'.

Line 441: How did you determine a "dominant" cluster? Is it above 50%? Above 20%? I think it would be good to add percentages in the text in brackets.

**Response:** We have added percentages in brackets to the text in order to support "dominant" clusters.

"The dominant soil clusters in each subregion differ from each other (Table 1). Clusters 3 (30%) and 1 (30%) are dominant in West Kunlun, clusters 2 (58%) and 7 (23%) in Gaize, clusters 7 (49%) and 1 (23%) in Aerjin, clusters 8 (55%) and 7 (18%) in G108, and cluster 8 (85%) in Wenquan."

Table 1. Area percentages of soil clusters in each subregion, over all subregions, and over the entire QTP. Unit: %

| Region | Cluster 1 | Cluster 2 | Cluster 3 | Cluster 4 | Cluster 5 | Cluster 6 | Cluster 7 | Cluster 8 |
|---|---|---|---|---|---|---|---|---|
| West Kunlun | 29.53 | 2.02 | 30.65 | 4.21 | 16.21 | 14.13 | 1.89 | 1.37 |
| Gaize | 1.11 | 57.75 | 5.34 | 12.62 | 0.12 | 0.44 | 22.55 | 0.06 |
| Aerjin | 22.97 | 12.39 | 8.51 | 3.88 | 0.40 | 2.88 | 48.76 | 0.20 |
| G308 | 6.24 | 9.56 | 4.40 | 7.29 | 0.00 | 0.12 | 17.24 | 55.15 |
| Wenquan | 7.27 | 6.37 | 0.72 | 1.62 | 0.00 | 0.04 | 9.01 | 74.97 |
| All subregions | 14.63 | 25.93 | 16.27 | 7.93 | 7.17 | 6.44 | 12.91 | 8.72 |
| QTP | 6.94 | 11.45 | 11.69 | 6.20 | 2.85 | 9.31 | 13.79 | 37.76 |

Lines 463 to 471: This section is really interesting to me. You could develop more on the potential links between the soil clusters (and characteristics) and permafrost occurrence. Then, you could contrast later in the text with the final map, which includes climatic factors, and highlight the importance of both when estimating permafrost occurrence. It's a suggestion.

**Response:**

Thanks for your suggestion. This point deserves further investigation. We planned to develop a new approach based on this strategy to study or isolate the contributions of climatic and environmental factors on permafrost occurrence over the QTP. It's really interesting. Thank you.

Line 497: You mention thermal erosion as an unfavorable condition leading to SFG. What do you mean exactly by thermal erosion? Thermal erosion is defined as the erosion of ice-rich permafrost by the combined thermal and mechanical action of moving water. Thus, it does occur in permafrost areas and is not symptomatic of non-permafrost environments. Do you actually mean heat advection by water, which is the transport of heat by water? This would make more sense as the heat from the large bodies of water would be enough to promote talik formation and thus prevent permafrost formation. Please explain if heat advection, and if so, change elsewhere in the text where there's mention of thermal erosion.

**Response:**

Thank you very much for pointing out this mistake that we made. We actually mean heat advection here. We have revised the text accordingly.

"SFG occurs extensively in the river source areas, namely the Three-River Headwaters Region, likely due to low latitude and the effects of heat advection by water flows that prevent permafrost formation."

We also check thoroughly the manuscript for the misuse.

Line 505: Consider changing the color scheme to avoid using blue as it is hard to distinguish the water bodies and the glaciers (see comment line 110). Also, please add the resolution of the map (1km).

**Response:**

The color scheme is now changed. Spatial resolution (1 km) has been added to the caption.

[Figure]

**Previous figure 9**

**Figure 8. Map of permafrost distribution at 1 km resolution over the QTP in 2010 (our map) produced in this study. Areas as well as percentages follow the legends. The map displays hill-shading with elevation.**

Line 555: "among which our map achieves the best performance with 54.5% accuracy in predicting SFG locations", does this affect the overall quality of the QTP map, especially considering that the western part is underrepresented in terms of boreholes?

**Response:**

There is a possible reason accounting for the low SFG accuracy: those boreholes are often drilled to

investigate permafrost status and those of seasonal frost were actually drilled in areas hard to distinguish. Few boreholes were drilled in obviously SFG areas unless for other purposes rather than for permafrost survey. This will lead to very few boreholes with SFG (e.g. only 11 out of 72 boreholes are in SFG), and these boreholes with SFG are often located near the boundary between permafrost and SFG, making them easy to be misidentified.

Therefore, we cannot verify the QTP permafrost maps directly on boreholes with SFG and relatively low accuracy in prediction SFG locations alone cannot indicate the overall quality of maps.

In our studies, we presented the accuracy in predicting SFG locations of three maps for comparison in which our map achieves the highest accuracy though the absolute value is not high. For assessing the overall accuracy, we rely on more performance metrics as well as cross-comparison with peer maps.

Line 570: I really like Figure 10 and I think it's one of the most important in the paper to highlight the performance of you map with regards to other works. A couple of things:

- You should change the color scheme according to your other maps based on my earlier suggestion
- You should enlarge the figure so that it is as big as possible, i.e., as large as the page
- I find it hard to compare one map with another. A possible solution to this would be to add a new class with "differences between modeled maps and the survey-based map". Seeing the highlighted differences would give a point of reference and help the reader identify the performance of each map with regards to the others. You did that for figures 11 and 12 and I think this figure would also benefit from it, but using the survey-based data as reference

**Response:**

1)Like previous figures, we changed the color of glaciers and lakes to make them more distinguishable but didn't change the color of permafrost because we believe a cold color like blue is more suitable to represent permafrost due to the fact of ground ice in permafrost body.

2)The figure has been enlarged.

3)Thanks for the suggestion. We tried and found boreholes information are very hard to distinguish when we overlay boreholes on the difference maps. Therefore, we decided to provide a new figure as a part of supplementary materials to show the differences between modelled maps and the survey-based map.

[Figure]

**Figure 9 (original fig 10). Spatial distributions of frozen ground in subregions from (a) survey-based maps, (b) our map, (c) the map from Zou et al. (2017), and (d) the map from Wang et al. (2019). Triangle symbols mark the locations of boreholes drilled around 2010 where the type of frozen ground was identified. Spatial differences between survey-based maps and three simulated maps from our study, Zou et al. (2017), and Wang et al. (2019) can be found in Figure S4 in supplementary materials.**

[Figure]

**Figure S4. Differences in spatial distribution of frozen ground type between survey-based maps and simulated**

maps from (a) our study, (b) Zou et al. (2017), and (c) Wang et al. (2019). "Both P" represents areas identified as underlying permafrost in both survey-based and simulated maps; "Both SFG" represents areas identified as seasonally frozen ground (SFG) in both survey-based and simulated maps; "Simulated-SFG and Survey-P" represents areas identified as underlying SFG in the simulated map but permafrost in survey-based maps; "Simulated-P and Survey-SFG" represents areas identified as underlying permafrost in simulated maps but SFG in survey-based maps.

Line 579: Table 5: This table could be condensed by having the rates in parentheses beside the confusion values. Change caption accordingly.

**Response:** Done.

Lines 587: This sentence needs to be rephrased because it sounds like your map is not that different from the Zou map, which defeats the purpose of this paper.

**Response:**

Thank you very much. We have rephrased the sentence to "The permafrost distributions in our map and Zou map are generally comparable, although ....".

Line 590: "Those headwater regions have been reported to be the critical regions where permafrost is more vulnerable and very sensitive to climate change (Jin et al., 2011; Zhang et al., 2021)." Could you expand on this and give more explanation as to why and why it is difficult to model?

**Response:**

These headwater regions are important because they are source areas of many major rivers in Asia, including the Yangtze River (the longest and largest river in China), the yellow River (the second longest river in China), the Mekong River (the longest and largest river in Southeast Asia) and many other important rivers. Permafrost dynamics, hydrological cycle and ecology in these headwater regions can affect vast downstream areas (Zhao et al., 2020).

Many studies have found that climate change has exerted profound impacts on the head water regions (Immerzeel et al., 2010; Jin et al., 2011; Zhang et al., 2021) and the permafrost there is characterized by high temperature (MAGT > -2.0 °C) and low thermal stability (Qin et al., 2017). One of our recent works show that the earliest permafrost degradation on the QTP under further global warming may occur in the head water regions (Zhang et al., 2022). Additional factors like vegetation dynamics and human activities (Zhang et al., 2016) could make the situation even more complex.

High temperature permafrost there makes it hard to be accurately distinguished from seasonally frozen ground. This requires high accuracy in modeling ground temperatures. More importantly, many local factors (terrain, vegetation, soil properties and so on) together determine the type of frozen soil, so

all of those factors must be accurately accounted for in a model.

We expanded the text according to the above explanations.

Refs:

Immerzeel, W.W., Van Beek, L.P. and Bierkens, M.F.: Climate change will affect the Asian water towers, Science, 328, 1382-1385, https://doi.org/10.1126/science.1183188, 2010.

Jin, H., Luo, D., Wang, S., Lü, L. and Wu, J.: Spatiotemporal variability of permafrost degradation on the Qinghai-Tibet Plateau, Sci. Cold Arid Reg., 3, 281-305, https://doi.org/10.3724/SP.J.1226.2011.00281, 2011.

Qin, Y., Wu, T., Zhao, L., Wu, X., Li, R., Xie, C., Pang, Q., Hu, G., Qiao, Y. and Zhao, G.: Numerical modeling of the active layer thickness and permafrost thermal state across Qinghai‑Tibetan Plateau, Journal of Geophysical Research: Atmospheres, 122, 11,604-11,620, https://doi.org/10.1002/2017JD026858, 2017.

Zhang, G., Nan, Z., Hu, N., Yin, Z., Zhao, L., Cheng, G. and Mu, C.: Qinghai‑Tibet Plateau permafrost at risk in the late 21st century, Earth's Future, https://doi.org/10.1029/2022EF002652, 2022.

Zhang, G., Nan, Z., Yin, Z. and Zhao, L.: Isolating the Contributions of Seasonal Climate Warming to Permafrost Thermal Responses Over the Qinghai-Tibet Plateau, Journal of Geophysical Research: Atmospheres, 126, https://doi.org/10.1029/2021JD035218, 2021.

Zhang, Y., Zhang, C., Wang, Z., Chen, Y., Gang, C., An, R. and Li, J.: Vegetation dynamics and its driving forces from climate change and human activities in the Three-River Source Region, China from 1982 to 2012, Sci. Total Environ., 563, 210-220, https://doi.org/10.1016/j.scitotenv.2016.03.223, 2016.

Zhao, L., Zou, D., Hu, G., Du, E., Pang, Q., Xiao, Y., Li, R., Sheng, Y., Wu, X. and Sun, Z.: Changing climate and the permafrost environment on the Qinghai-Tibet (Xizang) plateau, Permafrost and Periglacial Processes, 31, 396-405, https://doi.org/10.1002/ppp.2056, 2020.

Line 593: Figure 11: Same as other figures, reconsider color scheme. The figure could be larger to fit the width of the page. Also, the sections with more inconsistencies could be inserts on the side or below the map, otherwise we can't really see well. If you only want to use the figures after, please show the map as it is with the same classes and legend (i.e., Both P, Both SFG, Zou-P, etc.) before showing other maps (like in figure 12). Caption: change "tremendous" to "significant"

**Response:**

We changed the color scheme like the previous figures. But we decided not to insert inset maps here to display areas with more inconsistencies because those areas actually have been presented in the following figures and discussed in detail. The objective here is to show a general picture of the inconsistency between the two maps. We also found there is no enough space to insert sufficiently large inset maps, otherwise, the main figure will be squeezed a lot. If inset maps are too small, it does not look

any better than the original.

The same classes and legends have been applied to other figures where this figure is referred to. In the caption, "tremendous" has been replaced with "significant".

Line 616: Do you mean permafrost degradation or permafrost absence caused by the presence of the rivers? Because rivers don't necessarily cause degradation, but they do affect distribution. In addition, I'm having a hard time linking the distribution of the rivers with the DDT/DDF. Could you add arrows or more detailed explanations in the text as to where they affect permafrost? Unless you're just saying that their presence could potentially lead to more degradation? If so, rephrase to say this more explicitly.

**Response:**

Thank you for correcting this. We have revised the sentence to clarify this point: "the presence of these rivers could potentially lead to greater degradation of permafrost due to thermal advection of water flows."

Line 621: Figure 12: Color schemes to revise based on previous comments. Caption: Remove "Elaborate", for (c), please add indicators (e.g., arrows , boxes) and a description to help us understand what you want to illustrate between the two maps, otherwise we don't understand why it's there. The last sentence could be condensed to "See Figure 11 for notation"

**Response:**

Color scheme has changed.

Black ellipses have been added to show the areas where the presence of rivers could potentially lead to more permafrost degradation.

[Figure]

Figure 11 (original fig. 12). Maps of the areas between Altun Mountains and Kunlun Mountains (region b in Fig. 10) showing (a) detailed spatial differences in permafrost distribution between our map and the Zou map, (b) the ratios of DDT-over-DDF, and (c) a satellite image covering this region from Google Earth. The box indicates the Aerjin survey area with the availability of a survey-based permafrost map. P (survey-based) and SFG (survey-based) represent permafrost/seasonally frozen ground, respectively, on the survey-based Aerjin subregion map. See Figure 10 for notion. Ellipses mark areas where ground thermal state may have been affected by water flows.

Line 638: Could you give us an indication of the warming in the area, e.g., the mean annual temperature in 2010 vs 2020? Even better would be borehole temperatures to know how close to thawing the ground was in 2010.

**Response:**

In the source area of the Yangtze River, there is a borehole named QTP15 (Zhao et al., 2021) which can provide long-term annual-average soil temperature observations from 2006-2018. We can see that soil temperatures at 3 m, 6 m, and 10 m all experienced an increasing trend. Soil temperature at 10 m depth increased from -1.1°C in 2006 to -0.6°C in 2018, indicating that permafrost there is very close to thawing. The QTP15 borehole has been labelled in figure 12 (figure 13 in the original manuscript) and the soil temperature observations in this borehole have been added to supplementary materials.

Table S2. Annual-average soil temperatures at three depths (3m, 6m, and 10m) in the borehole QTP15 (33.10°N, 91.90°E) within the source area of the Yangtze River. Data source: Zhao et al. (2021). Symbol "/" denotes

**missing value.**

| Year | Soil temperature at 3m (°C) | Soil temperature at 6m (°C) | Soil temperature at 10m (°C) |
|------|------|------|------|
| 2006 | / | / | -1.1 |
| 2007 | -1.1 | -1.2 | -1.1 |
| 2008 | / | / | -1.2 |
| 2009 | -1.1 | -1.1 | -1.1 |
| 2010 | -0.8 | -1 | -1 |
| 2011 | -0.8 | -0.9 | -0.9 |
| 2012 | -0.7 | -0.8 | -0.8 |
| 2013 | -0.8 | -0.8 | -0.8 |
| 2014 | / | / | -0.8 |
| 2015 | -0.6 | -0.7 | -0.7 |
| 2016 | -0.5 | -0.6 | -0.7 |
| 2017 | -0.5 | -0.6 | -0.7 |
| 2018 | -0.7 | -0.8 | -0.6 |

Ref:

Zhao, L., Zou, D., Hu, G., Wu, T., Du, E., Liu, G., Xiao, Y., Li, R., Pang, Q. and Qiao, Y.: A synthesis dataset of permafrost thermal state for the Qinghai–Tibet (Xizang) Plateau, China, Earth Syst. Sci. Data, 13, 4207-4218, https://doi.org/10.5194/essd-13-4207-2021, 2021.

Line 639: Why does permafrost thawing occur more downstream than upstream? Is it because of elevation? If so, please explicitly say.

**Response:**

We clarified it. "In these areas, the occurrence of permafrost degradation usually recedes to upstream areas with higher elevations and cooler air temperatures."

Line 646: "likely statistically unreasonable", so is it or is it not? Not clear with the wording.

**Response:** 'likely' has been removed.

Line 651: Can you develop more as to why it is a bad thing that permafrost and SFG are overlapping for the Zou map and a good thing that your values are not overlapping? The entire paragraph (lines 643 to 652) looks more like just results than discussion, it needs to be developed

**Response:**

We have added more discussion about links between permafrost zonation index (PZI) and permafrost distribution to this paragraph, to explain why it is a bad thing in Zou map that PZI values in SFG regions overlap those in permafrost regions:

We further examined the two maps in the Yangtze River source areas using a PZI approach (Cao et al., 2019b). By definition, permafrost regions should have higher PZI values than SFG regions. The PZI map (Fig. 12d) used here was compiled based on 1475 in situ observations (Cao et al., 2019b), many of which were obtained between 2005-2018 in the vicinity of the G109 National Highway traversing the Yangtze River headwaters, making the PZI map a possible reference in this region. Therefore, we calculated PZI statistics in the permafrost and SFG zones of this region in our map and the Zou map (Fig. 错误!未找到引用源。e) to check the consistency with the PZI map in terms of permafrost distribution. The PZI statistics for permafrost in our map are close to those in the Zou map. However, for the PZI statistics in SFG regions, the lower and upper quartiles in the Zou map are 0.36 and 0.66, respectively, while the values in our map are 0.34 and 0.53, respectively, which means the SFG regions shown in our map have lower PZI values and are thus more suitable. The upper quartile for SFG regions (0.66) in the Zou map surpasses the lower quartile for permafrost regions, which is 0.55. The overlap is questionable because it suggests that some SFG regions have higher PZI values than permafrost regions in the same map, which contradicts the PZI definition. In contrast, the PZI intervals for both frozen ground types are more clearly distinguishable in our map.

Refs:

Cao, B., Zhang, T., Wu, Q., Sheng, Y., Zhao, L. and Zou, D.: Permafrost zonation index map and statistics over the Qinghai-Tibet Plateau based on field evidence, Permafrost and Periglacial Processes, 30, 178-194, https://doi.org/10.1002/ppp.2006, 2019b.

Gruber, S.: Derivation and analysis of a high-resolution estimate of global permafrost zonation, The Cryosphere, 6, 221-233, https://doi.org/10.5194/tc-6-221-2012, 2012.

Line 654: Figure 13:

• I suggest you put the same map (but zoomed in) as Figure 11, otherwise it's very difficult to see the differences between the two maps a) and b).

• Change the first sentence to have " (c) permafrost zonation index (PZI) in this region showing the spatial distribution of PZI"

• There's no mention of figure 13c in the text, and I don't really see why you have the figure here. You need to explain what it shows and mention it in the text.

- Explain what the red box is in the caption, not just in the text.
- Please a different notation (maybe dashed line) to circle the borehole from 2010 because it is confusing with the boreholes inconsistent with map
- Explanation for (d) needs to be repeated.

**Response**:

(1) We have added a map displaying the spatial differences between our map and the Zou map in the figure.

(2) The caption has been revised according to your comment.

(3) We have added a reference to figure 12d (figure 13c in the original manuscript) in the text. Through figure 13c, we want to show the spatial distribution of PZI and the locations of the Nation Highway G109 along which many ground data used to compile PZI maps were obtained (Cao et al., 2019b).

4) The explanation of the red box has been added to the caption.

(5) We removed the black circle notation in the figure. Instead using other notations, we distinguish the boreholes from different sources by labels: boreholes labelled by names are from Zhao et al. (2021), the others are from (Li et al., 2022). The caption has been revised as suggested.

(6) Explanation for subplot (e) (subplot (d) in the original figure) in the caption is added.

[Figure]

Figure 12 (original fig.13). Maps in the Yangtze River headwaters (region c in Fig. 10) showing permafrost distributions in (a) our map and (b) Zou map and (c) their spatial differences, along with (d-e) spatial and statistical permafrost zonation index (PZI) distributions in this region. The boreholes QTP11, QTP15 and TGLGT in (a) and (b) were drilled before 2010 and provided by Zhao et al. (2021), while the others were drilled in 2020 during the Second Tibetan Plateau Scientific Expedition (Li et al., 2022). The red box covers two boreholes of particular concern. Both boreholes at an elevation of 4870 m a.s.l. are

within a permafrost zone in both our map and Zou map, but revealed seasonal frost in 2020. For (c), the same notations apply as in Figure 10. (e) Box plot showing the statistical distributions of PZI values for permafrost and SFG regions in our map (Our-P and Our-SFG) and those in the Zou map (Zou-P and Zou-SFG). The center line in the box shows the median, the box shows the lower and upper quartiles, and the whiskers extend to the minimum and maximum data values.

Refs:

Cao, B., Zhang, T., Wu, Q., Sheng, Y., Zhao, L. and Zou, D.: Permafrost zonation index map and statistics over the Qinghai-Tibet Plateau based on field evidence, Permafrost and Periglacial Processes, 30, 178-194, https://doi.org/10.1002/ppp.2006, 2019b.

Li, J., Sheng, Y., Wu, J., Feng, Z., Ning, Z., Hu, X. and Zhang, X.: Landform-related permafrost characteristics in the source area of the Yellow River, eastern Qinghai-Tibet Plateau, Geomorphology, 269, 104-111, https://doi.org/10.1016/j.geomorph.2016.06.024, 2016.

Zhao, L., Zou, D., Hu, G., Wu, T., Du, E., Liu, G., Xiao, Y., Li, R., Pang, Q. and Qiao, Y.: A synthesis dataset of permafrost thermal state for the Qinghai‐Tibet (Xizang) Plateau, China, Earth Syst. Sci. Data, 13, 4207-4218, https://doi.org/10.5194/essd-13-4207-2021, 2021.

Line 660: Can you mention the year of the recorded data by the boreholes?

**Response:**

The recorded data by boreholes were obtained in 2013 and 2014, we have added this information to the text.

Line 675: Figure 14:
- Same comment as figure 13, I suggest you put the same map (but zoomed in) as Figure 11, otherwise it's very difficult to see the differences between the two maps a) and b).
- There needs to be mention in the caption as to why the 4300m is present, both in figure a-b but also in c.
- Explanation for (d) can refer to figure 13 or be repeated.

**Response:**

Revised as suggested. We have added map displaying the spatial differences between our map and the Zou map in the figure. The caption has been revised to mention that the lower limit of permafrost occurrence in this region is reported to be around 4300 m (Li et al., 2016) thus we present the contour as a reference to analyze the permafrost distribution.

[Figure]

Figure 13 (original fig.14). Maps in the Yellow River headwaters (region d in Fig. 10) showing permafrost distributions in (a) our map and (b) the Zou map and (c) their spatial differences, along with (d-e) spatial and statistical elevation distributions in this region. The contour of 4300 m as the lower limit of permafrost occurrence in this region a.s.l. is shown in (a), (b), and (d). (e) Box plot showing the statistical distribution of elevations in the permafrost and SFG zones in both maps. The same notations apply as in Figure 12.

Line 679: Data availability: I don't know if it's a journal requirement, but this section should go after the conclusion in my opinion.

**Response:**

It's a journal requirement to have a data availability section before the conclusion section.

**Technical corrections:**

Line 43: Add "the" in front of carbon cycle and change "thermodynamic" to "heat exchange"

Line 44: Begin sentence with "In addition"

Line 45: Remove sentence "Meanwhile, the consequences will lead to vital feedbacks to climate systems (Zhang et al., 2020; Wang et al., 2021)."

Line 52: You could add the references (Zhang et al., 2020; Wang et al., 2021)

Line 53: Join the two sentences together with "[…] for these studies because compared with the large […]"

Line 54: Change sentence to "Hence, there is a need for an accurate permafrost distribution map that would serve as a reference to validate results. The map could be used as a target to calibrate modeling parameters and to provide a constraint for future projection studies to minimize biases arising from the modeling process. Moreover, an accurate permafrost distribution map could serve as a fundamental

dataset for hydrological, carbon, ecological and engineering studies in cold regions (Hu et al., 2019; Li et al., 2020; Song et al., 2020; Mu et al., 2020)."

Line 62: Remove "the quality of these maps is often unsatisfactory, and"

**Response:**

All done as suggested.

Line 70: I don't like the use of "unsatisfactory" (see the related specific comment), and it too broad. Why is it unsatisfactory? Here you could remove "and unsatisfactory performances on the QTP" or change it to something like "their coarse spatial resolution inadequate for the QTP"

**Response:**

Thanks for pointing this out. We realized that 'unsatisfactory' is sometimes ambiguous. So here we replaced 'unsatisfactory performances' with 'lack of accuracy'.

Line 71: Verb tense? It looks like it should be "[…] would consequently restrict […]"

Line 73: Change "popular" to "common"

Line 74: Change "it will" to "Consequently, this leads to misrepresentation […]"

Line 79: Add "[…] to a large region with more spatial variability and thus more complex conditions."

Line 88: Change "firstly, secondly, last but not least" to "should be: 1) a map based on […]; 2) a map based on multi […]; and 3) a map of adequate […]"

Line 91: Remove "adequate accuracy", it's too vague, change the sentence to "a map that considers the impacts of local factors and that is well constrained during the mapping process"

Line 92: Change "under such circumstances" to "Based on these criteria"

Line 92: Change "provide" to "produce"

Line 92: Remove "high-quality"

Line 93: Change "over" to "of"

Line 94: Remove "fully"

Line 95: Change the sentence to "Our goal is also to provide a new reference map of 2010 for permafrost simulation studies of the QTP and provide a benchmark for transient land surface models under climate change"

Line 102: Remove "west to east in"

Line 111: Remove "(West Kunlun, Gaize, Aerjin, G308, and Wenquan, from west to east)"

Line 113: Remove "(WQ)"

Line 115: Cut sentence in two, e.g., "The land surface temperature (LST) product from the Moderate Resolution Imaging Spectroradiometer (MODIS) onboard the Terra and Aqua satellites is one of the most widely used LST products due to its high spatial and temporal resolutions (Wan, 2008). It has a global

coverage and has been applied in many permafrost mapping studies to provide temperature conditions (Gisnås et al., 2017; Zou et al., 2017; Obu et al., 2019; Wang et al., 2019)."

Line 120: Change "data" to "observations" and "are" to "were"

Line 123: Make into one paragraph with the previous one

Line 123: Remove "Moreover"

Line 123: Change "it is necessary" to "it was necessary"

Line 124: Change the sentence to: "For this reason, we used MODIS LST products from 2005, when automatic weather stations were put into operation in the study area, to 2010"

Line 123: Change "related to" to "influencing"

**Response:**

All done as suggested.

Line 130: Please reference the section to where vegetation was used to estimate DDT (section 3.2?)

**Response:**

We have added reference to section 3.2 in the paper as well as in the section S2 in supplementary materials.

Line 132: "to a spatial resolution of 1 km", could you add why? i.e., "to match the resolution of the LST data."

Line 149: The three paragraphs could be together in one paragraph since they are so short.

Line 149: Change "collected" to "used" and "revealed by" to "from"

Line 150: Remove "A newly published synthesis dataset of permafrost thermal state on the QTP" and change the sentence accordingly, it's too long and you can only cite the reference, no need to add more.

Line 152: Write 65 in letters because of beginning of sentence

Line 156: Numbers below 10 are usually written in letters, change accordingly in the rest of the text.

**Response:**

All done as suggested. Thank you for sharing us these rules. Really appreciate it.

Line 157: Change "are" to "were". There are multiple inconsistent verb tenses in the text, please verify throughout.

**Response:**

Done. We have checked and corrected the incorrect tenses throughout.

Line 161: Remove "during the Second Tibetan Plateau Scientific Expedition and Research"

Line 163: Change "at" by "in"

Line 166: Change title to "Comparison with existing QTP permafrost maps"

Line 168: Replace "cited in this study" with "used"

Line 168: Break the two maps into two sentences, too long

Line 175: Remove "Recently" and change "was" to "has been"

Line 177: Change "Cao et al. (2019a) regarded the Zou map as the best" to "Cao et al. (2019a) have determined that the Zou map is the best"

Line 179: Change "For the sake of simplicity" to "For simplicity"

Line 179 to 182: Long and difficult to read, to re-write, consider removing the name of the datasets and only cite the authors.

Line 185: Change "we applied a newly developed mapping method, namely FROSTNUM/COP (Hu et al., 2020)" to "we applied a mapping method developed by Hu et al. (2020)."

Line 185 to 190: Repetition of "this method", redundant

**Response:**

All done as suggested.

Line 191: Change DDT and DDF in the equation for Thawing and Freezing indices acronyms as mentioned in the specific comment about line 121.

**Response:**

Here, we have not made change because we want to keep DDT and DDF to respect the notations used in the paper proposing the frost number model (Nelson and Outcalt, 1987).

Line 207 to 215: Remove the verbs at the beginning of each description, e.g., "lists the, shows the, shows the, etc."

Line 223: Unless you still don't know, remove "theoretically"

**Response:**

All done as suggested.

Line 225: Rephrase "making it advanced in uncertainty control"

**Response:**

Rephrased it as "Moreover, while other all-weather LST products rely on multiple data resources with varying levels of uncertainty, this interpolation approach relies only on MODIS family data and thus has the advantage of controlling uncertainties during interpolation."

Line 226: Change the beginning of sentence to "First, in this SCSG-based stepwise interpolation

approach, clear-sky […]"

Line 247: Change to "the growing season"

Line 248: Change sentence to "We estimated the annual DDT from raw LST-derived thawing degree-days based on a multilinear regression model where GST is a function of independent variables including LST, NDVI, and latitude at weather stations Huang et al., 2020)."

Line 280: Remove "in this study", repetitive with the one said at the beginning of sentence

Line 290: Remove NDVI and FSC, then say "to which we added NDVI and FSC to further account […]"

Line 313: Remove "the" in "the types of frozen"

**Response:**

All done as suggested.

Line 313: What do you mean by neighborhood? Do you mean neighboring cells or surrounding cells? If so, change it and other mention of it.

**Response:**

We meant the neighboring cells, and we corrected it accordingly.

Line 367: Change 5 to five

**Response:**

Done.

Line 374: The sentence beginning with "in some regions" seems to be missing a word between QTP and thermally.

**Response:**

The sentence has been revised to: "In some regions of the QTP where permafrost is thermally controlled by elevation……"

Line 394: Remove "which generally underestimates the in situ annual DDT at most of the QTP weather stations from 2005 to 2010".

Line 395 to 397: Long sentence, to split into two.

Line 409: "up to 9000°C day" and "up to 8000°C day", Before you say "wide spectrum" so I was expecting a range, otherwise it's difficult to tell if you are saying the maximum range or the maximum valu of the range. Change the sentence accordingly.

**Response:**

We added the ranges "(from 0 to 9000 °C·day)" and "(from 0 to 8000 °C·day)".

Line 430: Figure 5: Change first sentence to "Bias correction of MODIS-LST-derived annual DDT with the interval-based approach". Change "from the diagonal line (black solid line)." to "from the 1:1 diagonal line (solid black line)."

Line 435: Figure 6: Enlarge the figures so that they fit the width of the page

**Response:**

All Done.

Line 440: "where lakes were excluded", do you mean in the subregions or in the approach? If you mean in the approach, rephrase.

**Response:**

We exclude lakes in the subregions so that the k-prototype approach wouldn't take them into consideration in spatial clustering and lakes did not belong to any soil cluster. We rephrased the sentence accordingly.

Line 444: Put at the beginning of section 4.2, maybe as the second sentence.

**Response:**

We move it to the beginning as suggested.

Line 445: Change "Figure 8 presents the primary characteristics of the soil clusters in the five subregions." to "In the five subregions, clusters 1, 2, and 3 […]".

Line 457: Remove "As summarized in Table 2" and add (table 2) at the end of the sentence.

**Response:**

All Done.

Line 461: Please specify what is the percentage covered by snow (ideally glaciers) from the study by Dail et al. (2018)

**Response:**

Dai et al. (2018) did not provide a precise area percentage of thick snow cover or glacier, nor did they provide their result dataset. However, Dai et al. (2018) show a map, from which most areas on the QTP are covered by relatively thin snow cover (<10cm). In this map, glaciers are not masked out. Many areas with thick snow cover overlap with glacier areas.

According to the 2nd glacier inventory (Guo et al. 2015), glacier cover about 1.55% of the whole QTP (Fig. R4) .

[Figure]

**Figure 8.** Spatial distribution of the average snow depth derived from AMSR-E and Moderate Resolution Imaging Spectroradiometer (MODIS) products using the dynamic algorithm for four seasons from 2003 to 2010 and lake distribution over the QTP.

Figure 8 from Dai et al. (2018)

[Figure]

Figure R4. Distribution of glaciers on the QTP during the period of 2006 to 2011. (Guo et al., 2015)

Ref:

Dai, L., Che, T., Xie, H. and Wu, X.: Estimation of snow depth over the Qinghai-Tibetan Plateau based on AMSR-E and MODIS data, Remote Sens., 10, 1989, https://doi.org/10.3390/rs10121989, 2018.

Guo, W., Liu, S., Xu, J., Wu, L., Shangguan, D., Yao, X., Wei, J., Bao, W., Yu, P. and Liu, Q.: The second Chinese glacier inventory: data, methods and results, J. Glaciol., 61, 357-372, https://doi.org/10.3189/2015JoG14J209, 2015.

Line 473: Figure 7: Please change the color scheme with colors that are less associated with temperature. Change figure 8 accordingly.

**Response:**

Color schemes were adjusted.

[Figure]

**Figure 6 (original fig.7). Resulting soil clusters in the five subregions and the predicted distribution of clusters throughout the QTP. A total of eight clusters were determined. Each soil cluster represents unique traits as reflected by a distinct value of model parameter E.**

[Figure]

**Figure 7 (original fig.8). Characteristics of the soil clusters in five subregions: (a) elevation; (b) slope; (c) normalized difference vegetation index (NDVI); (d) fractional snow cover (FSC); (e) topographic wetness index (TWI); (f) precipitation. All clusters are shown in different colors in correspondence with those in Fig. 6. The center line in the box shows the median, the box shows the lower and upper quartiles, and the whiskers extend to the minimum and maximum data values.**

Line 478: Figure 8: add to the figure that the values are obtained from the soil clusters in the five subregions, otherwise it looks like it's from the entire QTP.

**Response:**

Revised as suggested. Now it reads, "Characteristics of the soil clusters in five subregions: (a) elevation; (b) slope; (c) normalized difference vegetation index (NDVI); (d) fractional snow cover (FSC); (e) topographic wetness index (TWI); (f) precipitation."

Line 484: Table 3. Change to "Ranges and mean values as the most optimal values of the soil parameter E associated with the 8 soil clusters. The results were obtained from 1000 optimization trials."

**Response:**

It has been revised as suggested.

Line 488: Remove "Figure 9 shows" and change the sentence accordingly. Verb tense should be in the past. Mention the date.

**Response:**

Revised as suggested. "The resulting permafrost distribution map of 2010 on the QTP (Fig. 8 (Fig.9 in the original manuscript)) has a spatial resolution of 1km, according to which permafrost covered about $1.086 \times 10^6$ km$^2$, or 41.17% of the QTP, while SFG occupied about $1.447 \times 10^6$ km$^2$, or 54.85% of the total QTP area. The non-frozen ground was about $2.24 \times 10^4$ km$^2$, and the rest consisted of glaciers (about $4.08 \times 10^4$ km$^2$) and lakes (about $4.17 \times 10^4$ km$^2$)"

Line 488: Remove the "about" in that sentence, you give precise numbers so there's no point.

**Response:**

As our map is at 1km spatial resolution, the area could be accurate to 1 km$^2$ ($0.000001 \times 10^6$ km). However, to avoid too much digits in the text, for example, we rounded the permafrost area to thousands km$^2$ ($0.001 \times 10^6$ km), thus, there is 'about'.

Line 502: Do you mean a small amount of frozen ground? Because I mainly see SFG on the southeastern part of the QTP. Also, what do you mean by southeastern periphery? Periphery = the outer limits, and I'm not sure how to identify this on the QTP.

**Response:**

We meant within the extent of the QTP, non-frozen ground (minimum daily GST > 0°C) only covered a small part (0.85%) which locates in the southern QTP as displayed in green in Fig. 9.

We have revised the sentence accordingly: "Only a small amount of non-frozen ground exists in the southern margin of the QTP."

Line 510: Remove "high"

Line 510: Remove "high"

Line 547: Remove "well"

Line 552: Change "and the resulting measures are listed in Table 5" to "(table 5)"

Line 553: Change sentence to "Our map shows good agreement (ðœ… = 0.43) with the borehole observations in terms of ð□œ… compared to the Zou map (ðœ… = 0.30) and Wang map (ðœ… = 0.14)"

Line 584: Remove "widely recognized performance" and change sentence accordingly

Line 587: Change "consistent" to "comparable"

Line 589: Change "remarkable" to "noticeable"

Line 606: Remove "it seems that much"

Line 607: Remove "obviously"

**Response:**

All done as suggested.

Line 610: "Despite […]", rephrase the sentence, hard to read

**Response:**

Original: "Despite this imprecision, it at least follows that permafrost in this large region is extremely thermally unstable."

Revised: "This reflects that permafrost in this region is extremely thermally unstable."

Line 617: Remove "based on limited evidence"

**Response:**

Done.

Line 633: "for the frozen soil type" please specify exactly what that is.

**Response:**

We have deleted the words 'for the frozen soil type' to avoid confusion. The sentence have been revised as "Boreholes QTP11, QTP15 and TGLGT (Fig. 12), collected from Zhao et al. (2021), were drilled before 2010, and the frozen ground types at these boreholes were correctly identified in both maps."

Line 648: Remove "it is found that"

Line 652: Change "turn out to be more distinguishable" to "did not overlap"

Line 667: Remove "obviously"

**Response:**

All done.

Line 670: This paragraph needs to be phrased differently, it's awkward to read, especially the first sentence up to "The improved performance of our map may benefit from the increased accuracy of model inputs, consideration of local factors, and full exploitation of the survey-based subregion permafrost map. Therefore, as a better estimation of permafrost"

**Response:**

This paragraph seems redundant rather than a discussion so we just delete this paragraph.

Line 685: Remove "(Hu et al., 2020)" and you should mention that you use a modified version of it
Line 688: Change "(relative error < 10%)" to "with a relative error <10%"

**Response:**

All done.

Line 695: "to multifaceted effects of low latitude and local factors" be more explicit and list more, also I'm not sure thermal erosion is appropriate (see earlier comment).

**Response:**

Our mapping approach is unable to explicitly determine which local factors cause the occurrence of seasonally frozen ground in the source areas of rivers. Therefore, we decided to remove the description of possible reasons from this sentence.

Line 697: Remove the Kappa coefficient and simply mention that your map performed better than other recently published maps. Same for the lines after, reduce the numbers and talk about main conclusions for the comparison part of the paper.
Lines 701 to 706: Sentence too long, cut into two sentences

**Response:**

All done.

Line 707: This does not need to be a paragraph on its own, change the initial The to "Our"

**Response:**

We revised as suggested and combined this paragraph with the previous paragraph.

Line 708: Remove "of sufficient quality"
Line 707: Change the sentence to "The new 2010 permafrost distribution map provides accurate and fundamental information for QTP permafrost and can serve as a historical reference when projecting future changes of QTP permafrost and as benchmark map to calibrate/validate spatial simulations of land

surface models"

**Response:**

Done as suggested.

---

## Author Response (AR2)

**Responses to the Reviewer**

Text in red are the reviewer's comments; **those in black** are the authors' replies and explanations to the reviewer's comments; and those in blue are the revised texts appeared in the revised manuscript.

First of all, we would like to thank Dr. Samuel Gagnon for providing invaluable comments and enhancing the quality of our manuscript. Below please find our responses to the comments:

Line 36: "complex local factors", could you give one or two examples?

Line 94: temperature: could you give the range and also the mean for a specific time period, e.g., 1990-2020? Could be something like "The mean annual air temperature for the 1990-2020 period ranges from -5 to 5°C (mean = -2.4°C).

Line 101: You mention that permafrost has formed extensively, but could you specify if it's continuous or extensive discontinuous?

**Response:**

Thank you for your valuable input. We have incorporated your suggestions by adding examples of "terrain, vegetation cover, soil properties, and hydrological conditions" as local factors. Additionally, we have specified the time period as "during the period 1981–2010" when discussing the temperature range. Lastly, we have revised the description of permafrost distribution as "Extensive alpine permafrost has formed across the QTP, featuring continuous permafrost in the central region and discontinuous permafrost in the southern parts (Yi et al., 2014)"

Refs:

Yi, S., Wang, X., Qin, Y., Xiang, B. and Ding, Y.: Responses of alpine grassland on Qinghai–Tibetan plateau to climate warming and permafrost degradation: a modeling perspective, Environ. Res. Lett., 9, 74014, https://doi.org/10.1088/1748-9326/9/7/074014, 2014.

Line 332-333: Do you not report DDF in negative values? According to your definition, they should be. You should either mention at the beginning (line 133) that DDF are in absolute values to facilitate comparisons with DDT, or change in the text for negative values (and change figures like Fig. 3) accordingly.

**Response:**

Thanks. To adhere to the definition provided in equation (1) where both DDT and DDF are placed within the square root, we define DDT and DDF as absolute values.

Line 311: As in my previous review, this type of sentence describing the figure should be avoided in

the text, I suggest you go directly in what you want to say, e.g., "In-situ DDT (DDF) was compared with raw-LST-derived DDT (DDF) to determine the fit of the model (Fig. 3)."

**Response:**

Yes, we understand and still avoid to rewrite it like describing methodology because this is in the section presenting the results. The revised text reads as follows:

"Figure 3 illustrates the comparison between average annual in situ DDT (DDF) values, calculated as averages over 2005-2010 from daily mean GSTs at 131 weather stations on the QTP, and the average annual satellite DDT (DDF) values at the corresponding MODIS pixels derived directly from daily mean MODIS LSTs. The raw LST-derived DDF values exhibited a perfect match with the in situ DDF values, echoing the limited effects of thin and short-duration snow cover on the thermal states of underlying soils on the QTP (Wu and Zhang, 2008; Zhao et al., 2017). "

It still begins with Figure 3 but we focus on its function as comparing the two sources of DDT/F values.

Line 403: Please add the percentage cover for non-frozen ground to be consistent with the rest and facilitate reading.

**Response:**

We have added the percentage covers: "The non-frozen ground was about $2.24 \times 10^4$ km$^2$ (0.85% of the QTP), and the rest consisted of glaciers (about $4.08 \times 10^4$ km$^2$, 1.55%) and lakes (about $4.17 \times 10^4$ km$^2$, 1.58%)."

Line 458: "showed fewer permafrost areas", is this compared to the local survey map or compared to the Wang map? Please specify.

**Response:**

We have specified it: "……, both the Zou map ($\kappa$ = 0.48) and our map ($\kappa$ = 0.68) showed fewer permafrost areas compared to the local survey map"

Line 465: Table 4, should it be Table 3?

**Response:**

It should be Table 4 here. In line 465, we evaluated maps by boreholes, the Cohen's Kappa between maps and boreholes were shown in the last row of Table 4, while Table 3 demonstrates the Cohen's Kappa between modelled maps and survey-based maps.

Line 635: "In typical regions with distinct differences", to rephrase, I'm not exactly sure what you mean.

**Response:**

The sentence has been rephrased as "In some regions where we found distinct differences between our map and Zou map".

Line 449: Could you add an example of "thermal perturbations"? E.g., "highly susceptible to thermal perturbations such as a warmer summer".

Line 56: remove "that require sufficient accuracy"

Line 92: The coordinates both indicate East, change one for North

Line 92: Change the sentence to "The QTP (bounded within 73.5–104.5°E and 26–40°N) is a high-elevation flat terrain of about 2.6×106 km2 surrounded by surrounded by high mountain ranges (Fig. 1)."

Line 95: remove "in most areas between 3000-5000 m a.s.l."

Line 96: Change "In the last five decades preceding 2010" to "From 1960 to 2010"

Line 97: Change to "Mean annual precipitations decrease from more than 700 mm in the southeast to about 50 mm in the northwest, and about 90% of precipitations fall during the growing season from May to September"

Line 99: Move this sentence to the beginning of the section, I suggest as the second sentence, right before "Most of the QTP lies between…"

Line 101: Change sentence to "Alpine permafrost has formed extensively on the QTP (ADD SOMETHING ABOUT CONTINUOUS/DISCONTINUOUS). Ice-rich layers are often found near the permafrost table, which is generally 2–3 m deep (Zhao et al., 2020)."

Line 158: Change times to "2:00, 8:00, 14:00, and 20:00" and remove o'clock

Line 165: could you list the three sources with first, second, third to facilitate reading? E.g., "First, a newly published […]", "Second, seven boreholes were collected […]", and "Third, in the Yangtze River […]"

Line 202: remove "where DDF and DDT represent annual ground surface freezing and thawing indices (°C·day), respectively", already defined earlier in the text, you could directly explain E.

Line 239: Change beginning of paragraph to "Conversely, vegetation cover affects DDT by providing a strong […]"

Line 265: Remove "Since there is no simple way to determine wk and wb"

Line 301: Remove "(Zou et al., 2017; Wang et al., 2019)", since you defined the names of the map from those references, you don't need to cite them every time (like in the next sentence). To correct everywhere.

Line 323: Could you replace "below the 10% level" with something like "below our accepted level of error (<10%)", otherwise this information is useless.

Line 422: End sentence after (Fig. 9). Then, start a new one "Our map had a Cohen's Kapa coefficient

of about […]"

Line 453: Change "For the Wang map" to "Since the Wang map"

Line 455: Change sentence to "Together, all of these factors caused the Wang map to overestimate permafrost extent in Gaize"

Line 479: Change decent to statisfactory

Line 501: Change "terrain, vegetation, soil properties and so on" to "e.g., terrain, vegetation, soil properties"

Line 522: Remove "especially"

Line 572: Change to "The red boxes in a) and b) cover two boreholes of […]"

Line 580: Change "Our map proved to be more accurate" to "Our map was more accurate"

Line 593: Change sentence to "Despite the better performance of our map compared to other available products, our mapping approach had limitations and left room for potential improvements"

Line 594: Change "leveraged" to "extracted"

**Response:**

All done as suggested. For more detailed revisions, please refer to the track-changes doc.